

# Study of radiative properties and effects of high-altitude cirrus clouds in Barcelona, Spain with 4 years of lidar measurements

Cristina Gil-Díaz[1], Michäel Sicard[1,2,3], Odran Sourdeval[4], Athulya Saiprakash[4], Constantino Muñoz-Porcar[1], Adolfo Comerón[1], Alejandro Rodríguez-Gómez[1], Daniel Camilo Fortunato dos Santos Oliveira[1]

[1]CommSensLab, Dept of Signal Theory and Communications, Universitat Politècncia de Catalunya (UPC), Barcelona, 08034, Spain

[2]Ciències i Tecnologies de l'Espai-Centre de Recerca de l'Aeronàutica i de l'Espai/Institut d'Estudis Espacials de Catalunya (CTE-CRAE/IEEC), Universitat Politècnica de Catalunya (UPC), Barcelona, 08034, Spain

[3]Laboratoire de l'Atmosphère et des Cyclones, Université de La Réunion, Saint Denis, 97744, France

[4]Laboratorie d'Optique Atmosphérique (LOA), Université de Lille, Villeneuve d'Ascq, 59650, France

**Correspondence:** Cristina Gil-Díaz (cristina.gil.diaz@upc.edu)

**Abstract.** Cloud-radiation interaction still drives large uncertainties in climate models and its estimation is key to make more accurate predictions. In this context, the high-altitude cirrus clouds play a fundamental role, because 1) they have a high occurrence frequency globally and 2) they are the only cloud that can readily cool or warm the atmosphere during daytime, depending on their properties. This study presents a comprehensive analysis of radiative properties and effects of cirrus clouds

based on 4 years of continuous ground-based lidar measurements with the Barcelona (Spain) Micro Pulse Lidar. First, we introduce a novel approach of a self-consistent scattering model for cirrus clouds to determine their radiative properties at different wavelengths using only the effective extinction coefficient and mid-cloud temperature. Second, we calculate the radiative effects of cirrus clouds with the Discrete Ordinates Method and we validate our results with SolRad-Net pyranometers and CERES measurements. Third, we present a case study analyzing the radiative effects of a cirrus cloud along its back-

trajectory using data from the Chemical LAgrangian Model of the Stratosphere with microphysics scheme for Ice clouds formation. The results show that the cirrus clouds with an average ice water content of $4.97\pm5.53$ mg/m$^3$, at nighttime, warm the atmosphere at top-of-the-atmosphere (TOA; +50.1 Wm$^{-2}$) almost twice than at bottom-of-the-atmosphere (BOA; +23.0 Wm$^{-2}$); at daytime, they generally cool the BOA (-8.57 Wm$^{-2}$, 80% of the cases) and always warm the TOA (+18.9 Wm$^{-2}$). In these simulations, the influence of the lower layer aerosols is negligible in the cirrus radiative effects, with a BIAS of -0.71%.

For the case study, the net radiative effects produced by the cirrus cloud, going at TOA from 0 to +42 Wm$^{-2}$ and at BOA from -51 to +20 Wm$^{-2}$. This study reveals that the complexity of the cirrus cloud radiative effect calculation lies in the fact that it is highly sensitive to the cirrus scene properties.



## 1    Introduction

Cloud-radiation interaction still drives large uncertainties in weather and climate models (IPCC, 2023). Its estimation is very important in order to understand the main physical processes driving climate change, to predict long-term global warming and to make more accurate weather predictions. (Loeb et al., 2009) estimated globally at top-of-the-atmosphere an annual cloud shortwave radiative effect of approximately -50 Wm$^{-2}$ and longwave effect of approximately +30 Wm$^{-2}$. The resulting net global mean cloud radiative effect of approximately -20 Wm$^{-2}$ implies a net cooling effect of clouds on the current climate. Owing tot he large magnitudes of the cloud radiative effects, clouds cause a significant climate feedback that depends on cloud properties and their spatial distribution (IPCC, 2023). In this context, the high-altitude cirrus clouds play a fundamental role in the global radiation budget (Liou, 1986; Lolli et al., 2017), having been designated as poorly understood by (IPCC, 2023) because of a lack of knowledge of their dynamic, microphysical and radiative properties. Indeed, cirrus cloud critical role in the climate comes from the fact that 1) they have a high occurrence frequency globally (Holz et al., 2008) and 2) they are the only cloud that can readily cool or warm the top-of-the-atmosphere and bottom-of-the-atmosphere, during daytime, depending on their properties (Campbell et al., 2016). In fact, (Campbell et al., 2016) demonstrated through a one-year long lidar dataset that positive or negative daytime cirrus cloud forcing could occur depending on the cloud optical depth (COD) and the solar zenith angle (SZA).

Cirrus clouds are mainly composed of ice crystals and can form through different atmospheric mechanisms that determine their initial properties and further evolution. In European field campaigns it has been observed that during a low or high pressure system, cirrus clouds are typically formed by a slow updraft, while in conjunction with jet streams or gravity waves, cirrus clouds originate as a consequence of a fast updraft. Also, liquid origin cirrus mostly related to warm conveyor belts are found (Kramer et al., 2016). The most common parameters that are measured in cirrus clouds are temperature, relative humidity (for ice), vertical velocity, ice water content, ice crystal number, ice nucleation particles and ice crystal size distribution. Unfortunately, the measurements of ice crystal number and size as well as relative humidity have faced instrumental problems during last decades (Korolev et al., 2011; Kramer et al., 2016). Moreover, it is a difficult task to draw conclusions about the microphysical processes of cirrus clouds from these observations. Nevertheless, worldwide studies on cloud and aerosol optical and microphysical properties have increased significantly over the last years through the passive ground-based measurements made e.g. by the European Aerosol Research LIdar NETwork, EARLINET (Pappalardo et al., 2014) now included in the Aerosols, Clouds and Trace gases Research Infrastructure, ACTRIS (Saponaro et al., 2019), Micro Pulse Lidar NETwork, MPLNET (Welton et al., 2001); and satellite measurements e.g. by Cloud-Aerosol Lidar and Infrared Pathfinder Satellite Observations, CALIPSO (Winker et al., 2007), AEOLUS (Ingmann and Straume, 2016), MODerate-resolution Imaging System, MODIS (Levy et al., 2013) and in the future, Earth Cloud, Aerosol and Radiation Explorer, EarthCARE (Eisinger et al, 2017). Additionally, in-situ airborne measurement campaigns have been carried out such as the First ISCCP Project Regional Experiment, FIRE from 1989 to 1995 (Ackerman et al., 1990; Heymsfield et al, 1990; Heymsfield and Miloshevich, 1995), the International Cirrus Experiment, ICE campaign in 1989 (Raschke et al., 1987), EUropean Cloud Radiation EXperiment, EUCREX in 1993





and 1994 (Sauvage et al., 1999), Field Radiation Experiment on Natural Cirrus and High-level clouds, FRENCH in 2001 (Brogniez et al., 2004), Tropical Composition, Cloud and Climate Coupling, TC4 campaign in 2007 (King et al., 2010; Toon et al., 2010) and CIRRUS-HL campaign in 2021, which is the follow-up to the CIRRUS-ML campaign in 2017 (Voigt et al., 2017; 55 De La Torre Castro et al., 2023).

Up to the present, there are three possibilities for characterising cirrus clouds. One option is the use of in-situ airborne measurements. A second option is to work with a microphysical cirrus cloud model like the Chemical LAgrangian Model of the Stratosphere with microphysics scheme for Ice clouds formation (CLaMS-Ice) (Spichtinger and Gierens, 2009) or Model 60 for aerosol and ice dynamics (MAID) (Bunz et al., 2008; Rolf et al., 2012), that simulate the cirrus cloud development based on the cirrus bulk model by along back-trajectories. The main advantage of this choice is that there is no need to have in-situ airborne lidar measurements. A third option is to employ lidar measurements for the characterisation of cirrus clouds. For that purpose, it is necessary to use a method such as the two-way transmittance method to characterize cirrus clouds optically (Gil-Díaz et al., 2024) together with a scattering model to obtain radiative retrievals. For example, (Baran and Labonnote, 2007; 65 Baran et al, 2009, 2011a, b, 2014) relates the cirrus ice water content (IWC) and mid-cloud temperature with its extinction coefficient, single scattering albedo (SSA) and asymmetry factor (asyF). Alternatively, (Heymsfield et al., 2014; Dolinar et al., 2022) propose to calculate the cirrus ice water content from the extinction coefficient at a visible wavelength and the effective geometric diameter of the ice crystals, which in turn is a function of temperature. Once the cirrus ice content and the effective geometric diameter of ice crystals are obtained, the scattering and absorption coefficients and the asymmetry factor can be 70 calculated with the (Fu et al., 1998, 1999) parametrizations.

The objective of this paper is to analyze the radiative properties and forcings of cirrus clouds based on 4 years of continuous ground-based lidar measurements obtained from the NASA Micropulse lidar network (MPLNET, https://mplnet.gsfc.nasa. gov/) in Barcelona. Specifically, the radiative properties of cirrus clouds have been calculated with a new approach of the self-75 consistent scattering model for cirrus clouds (Baran and Labonnote, 2007; Baran et al, 2014), using only lidar measurements and radiosounding data and their radiative forcings have been calculated with the ARTDECO package. The instrumentation used is presented in Section 2. A new approach of the self-consistent scattering model for cirrus clouds, the radiative transfer model DISORT and the CLaMS-Ice model are presented in Section 3.1, Section 3.2 and Section 3.3, respectively. The results obtained in this paper are shown in Section 4 and conclusions are presented in Section 5.

## 80  2   Instrumentation

The radiative characterization of cirrus clouds relies on the results obtained from (Gil-Díaz et al., 2024) and the instrumentation detailed below.



## 2.1  NASA Micro-Pulse Lidar Network

A more detailed description of this instrumentation can be found in (Gil-Díaz et al., 2024). In this study, we use the Aerosol
(AER) product, provided at 1-min temporal resolution and at 75m vertical resolution to characterize the aerosol layer which is
closest to the surface. The MPLNET AER product includes sun/lunar photometer observations to invert lidar signal to obtain
aerosol properties during 24h at day. Aerosol extinction, backscatter and the column lidar ratio among other properties belong
to the MPLNET AER product (Welton et al., 2000, 2002).

## 2.2  Meteorological Service of Catalonia

Radiosondes are launched twice every day (at 00:00 and 12:00 UTC) by the Meteorological Service of Catalonia (Meteocat)
at a distance of less than 1 km from the MPL site. The radiosondes provide measurements of pressure, altitude, temperature,
relative humidity, wind speed and direction. Only altitude, pressure and temperature profiles are used in the present work.

## 2.3  Solar Radiation Network

Solar Radiation Network (SolRad-Net, https://solrad-net.gsfc.nasa.gov/) is a federation of ground-based sensors providing
high-frequency solar flux measurements in quasi-real time to the scientific community. This program was implemented as
a companion to AERONET and its instrumentation is collocated in the AERONET sites. Each SolRad-Net site is initially
equipped with two flux sensors: a Kipp and Zonen CM-21 pyranometer (0.305-2.8 $\mu$m) for measuring the total solar spectrum
and Skye Instruments SKE-510 PAR (photosynthetically-active radiation) Energy sensor (spectral range: 0.4-0.7 $\mu$m).

In this study, measurement from the Kipp & Zonen instrument is employed with the Level 1.0, corresponding to data
unscreened that do not have final calibration applied, during the period from 2018 to 2022. The Kipp & Zonen CM-21 units are
ISO 9060 Secondary Standard thermopile pyranometers featuring a receiving element housed beneath two concentric Schott
K5 glass domes. More information about the instrument can be found at the following web link (https://www.kippzonen.com/
Product/14/CMP21-Pyranometer).

## 105  2.4  Clouds and the Earth's Radiant Energy System project

The Clouds and the Earth's Radiant Energy System project (CERES, https://ceres.larc.nasa.gov/) provides observations of
Earth's radiation budget using measurements from CERES instruments onboard the Terra, Aqua and Suomi National Polar-
orbiting Partnership (S-NPP) and NOAA-20 (formerly JPSS-1, named after the Joint Polar Satellite System-1 mission) satellites
(Loeb et al., 2016). The goals of the CERES project are: (1) to produce a long-term and integrated global climate data record
for detecting decadal changes in the Earth's radiation budget from the surface to the top-of-the-atmosphere; (2) to improve
understanding of how Earth's radiation budget varies temporally and spatially and the role that clouds and other atmospheric
properties play; (3) to support climate model evaluation and improvement through model-observation intercomparisons.



On one hand, the dataset processed is CERES instantaneous Single Scanner Footprint (SSF) product with the Level 2.
Specifically, the product analysed is the observed top-of-the-atmosphere upward fluxes and the surface emissivity in longwave spectral range (5-35 $\mu$m), during the period which covers from 2018 to 2022. The surface albedo is calculated as the unity minus the surface emissivity. The CERES measurements are either from the AQUA satellite (overpass over Barcelona between 12:00 and 13:00 UTC) or from the TERRA satellite (overpass over Barcelona between 10:00 and 10:30 UTC). This variable is provided with a surface spatial resolution of 20 km at nadir (Su et al., 2015).

On the other hand, Synoptic TOA and surface fluxes and clouds (SYN) product with the Level 3 is processed to get a value for albedo over the Atlantic Ocean and the Mediterranean Sea in order to study the radiative forcing produced by a cirrus cloud along its back-trajectory. Specifically, the variable analysed is the surface albedo from the SYN1deg product, which is integrated between 0.25-4 $\mu$m, with a daily temporal resolution and a spatial resolution of 1°x1°.

## 2.5   NASA AErosol RObotic NETwork

The NASA AErosol RObotic NETwork (AERONET, https://aeronet.gsfc.nasa.gov/) is a federation of ground-based sun/lunar-photometers established by NASA and LOA-PHOTONS (CNRS). For more than two decades, the project has provided long-term, continuous, and readily accessible public domain database of aerosol optical, microphysical and radiative properties for aerosol characterization research and validation of satellite retrievals, and synergism with other databases. The network imposes standardization of instruments, calibration, processing and distribution.

In this work, Version 3.0 and Level 1.5 (cloud-screened and quality-controlled) inversion products are used for the time period from 2018 to 2022. In order to characterize the aerosols in the lowest layer of the troposphere in the shortwave spectrum (SW, 0.2-4 $\mu$m), variables like Aerosol Optical Depth (AOD), Single Scattering Albedo (SSA) and Asymmetry Factor (asyF) are used. For other wavelengths than the working wavelengths of AERONET, in the SW range, the Angström exponent is used (Wagner and Silva, 2008). For example, the Angström exponent is calculated with the AOD values at 440 and 675 nm to obtain the AOD at 550 nm. For the characterization of the aerosol layer in the longwave spectrum (LW, 2-40 $\mu$m), the optical properties mentioned above are extracted from the Laboratory for Information Technologies and Mathematical Simulation (LITMS) database (Rublev et al., 1994).

## 2.6   The Cloud-Aerosol Lidar and Infrared Pathfinder Satellite Observation

A more detailed description of this instrumentation can be found in (Gil-Díaz et al., 2024). In order to validate the product ice water content from the Lagrangian microphysical cirrus model CLaMS-Ice (see Section 3.3), the CALIPSO product used is the "5 km Cloud Layer (05kmCLay)" product, with the Level 2 (L2) and Version 4.20 (V4.20), available from June 2006. This product has a 5-km horizontal averaging resolution with a maximum of ten layers reported per profile. In particular, this product contains geometrical, thermal and optical properties of each cloud layer detected like layer top/base altitude and temperature, integrated attenuated backscattering coefficient at 532 nm and 1064 nm, integrated particle depolarization ratio or



ice water path. The ice water content coefficient of the cloud layer is estimated as the ratio between the ice water path and the geometrical thickness of the layer.

## 3 Methodology

The radiative properties of cirrus scenes are determined through the use of a self-consistent scattering model for cirrus clouds (Baran and Labonnote, 2007; Baran et al, 2014) and their radiative forcings are calculated with the ARTDECO package, which implements a variety of optical properties into state of the art radiative transfer models (see below).

### 3.1 The self-consistent scattering model for cirrus clouds

The self-consistent scattering model for cirrus clouds consists of an ensemble of six ice crystal members, where the simplest ice
crystals are represented by hexagonal ice columns with an aspect ratio of unity and the more complex ice crystals are formed by arbitrarily and randomly oriented attaching other hexagonal elements, until a chain-like ice crystal is formed. The complexity of ice crystals tends to increase with increasing size, and they generally become more spatial, with the hexagonal components becoming more elongated (Heymsfield and Miloshevich, 2003). The geometrical configuration consists in an ensemble of six members, the first of which is the hexagonal ice column, the second the six-branched bullet rosette; thereafter, hexagonal
monomers are arbitrarily attached to each other, as a function of the maximum dimension, forming three- to ten-element hexagonal ice aggregates. This ensemble tries to represent ice crystals of cirrus clouds observed in different measurement campaigns (Heymsfield and Miloshevich, 2003; Lawson et al., 2003; Connolly et al., 2005). For example, bullet-rosettes are included in the ensemble as these are commonly observed in mid-latitude and arctic regions (Lawson et al., 2006; Schmitt et al., 2006) and its geometry is described in (Macke et al., 1996). The ensemble members are constructed so as not to contain intersect-
ing planes, and the crystals are attached such that multiple reflections between them are negligible, which was determined experimentally using ray-tracing calculations. The first element represents the smaller sizes of ice crystals in the particle size distribution (PSD), whilst the hexagonal ice aggregates represent the process of ice crystal aggregation and thus represent the larger sizes of ice crystals in the PSD. The members of the ensemble are distributed into six equal intervals of the PSD.

The PSD assumed is the (Field et al., 2007) moment estimation parametrization of the PSD, referred to as F07, which relates the decimal logarithm of ice water content (LIWC) to a radiative property (scattering, absorption coefficient and asymmetry factor) via a polynomial fit to the in-cloud temperature (T), with a spectral dependence, as is shown in the following expressions:

$$\log_{10} \sigma_{ef,\lambda}(z,t) = c_{ef,\lambda,1} + c_{ef,\lambda,2} \, T(z,t) + c_{ef,\lambda,3} \, LIWC(z,t) + c_{ef,\lambda,4} \, T^2(z,t) + c_{ef,\lambda,5} \, LIWC^2(z,t)$$
$$+ c_{ef,\lambda,6} \, T(z,t) \, LIWC(z,t) \tag{1}$$

$$asyF_\lambda(z,t) = a_{\lambda,1} + a_{\lambda,2} \, T(z,t) + a_{\lambda,3} \, LIWC(z,t) \tag{2}$$



Where the coefficients depending of the degree (n) and wavelength ($\lambda$) are different for the different radiative effects $\sigma_{ef,\lambda} \equiv \sigma_{sc,\lambda}, \sigma_{abs,\lambda}, asyF_\lambda$ scattering, absorption coefficients and asymmetry factor, respectively. The units of the variables are: T [K], LIWC [kg/m$^3$] and $\sigma_{ef,\lambda}$ [m$^{-1}$]. The parametrization represented by Eq. 1, 2 is based on 10000 in-situ measurements of the PSD, obtained in tropical and mid-latitude cirrus, at temperatures between -60ºC and 0ºC. The F07 parametrization ignored in-situ measurements of ice crystal size less than 100 $\mu$m, due to the problem of ice crystal shattering on the inlet of closed-path instruments (Korolev et al., 2011). Therefore, for these ice crystal sizes, F07 assumed an exponential fit to the PSD. Conversely, for crystal size greater than 100 $\mu$m, filtering was applied to the measured PSDs to reduce the likelihood of shattered ice crystal artefacts being included in the parametrization.

In this study, a new methodology for the calculation of the radiative properties of cirrus clouds at different wavelengths is proposed, as shown in Fig. 1. First, the IWC of the cirrus cloud that has no spectral dependence is calculated by introducing in Eq. 3 the effective column extinction coefficient calculated with the two-way transmittance method at 0.532 $\mu$m (the working wavelength of the MPL), for a mid-cloud temperature. Eq. 3 is derived from Eq. 1, where $\sigma_{ef,\lambda} = \sigma_{ext,\lambda_o}$ is assumed. Thus, a simple way of calculating the IWC from an extinction coefficient and a cloud temperature has been implemented by assuming the absence of absorption, which is entirely reasonable because the wavelength used belongs to the visible spectral range (Sun and Shine, 1994).

$$LIWC(z,t) = \frac{-(c_{sc,\lambda_o,3} + c_{sc,\lambda_o,6}\, T(z,t)) \pm \sqrt{(c_{sc,\lambda_o,3} + c_{sc,\lambda_o,6}\, T(z,t))^2 + 4\, c_{sc,\lambda_o,5}\, \Theta(z,t)}}{2\, c_{sc,\lambda_o,5}} \qquad \lambda_o \in [0.38, 0.7]\,\mu m$$

$$\Theta(z,t) = \log_{10} \sigma_{ext,\lambda_o}(z,t) - c_{sc,\lambda_o,1} - c_{sc,\lambda_o,2}\, T(z,t) - c_{sc,\lambda_o,4}\, T^2(z,t)$$

(3)

The working wavelength of the lidar system is defined as $\lambda_o$ and the extinction retrieved by the lidar system is denoted as $\sigma_{ext,\lambda_o}$. Once, the IWC of the cirrus cloud is calculated, this variable is introduced into Eq. 1, 2 to get the absorption, scattering and asymmetry factor coefficients for each wavelength. From these variables, the extinction and single scattering albedo are also calculated for each wavelength. This method is valid for any lidar working wavelength which belongs to the visible spectrum.

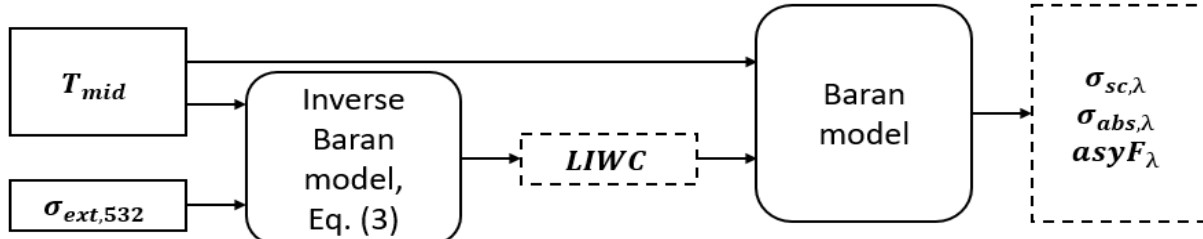

**Figure 1.** Scheme of the new model approach of the self-consistent scattering model for cirrus clouds.





The main advantage of this model is that it is not necessary to calculate the ice particle shapes and their size distributions
in order to calculate their radiative properties. These variables can be obtained easily with only elastic lidar systems and
radiosondes or meteorological models. An example of the application of this method is shown in Fig. 2, for a cirrus cloud of
08/12/2018 at 12 UTC, measured in Barcelona.

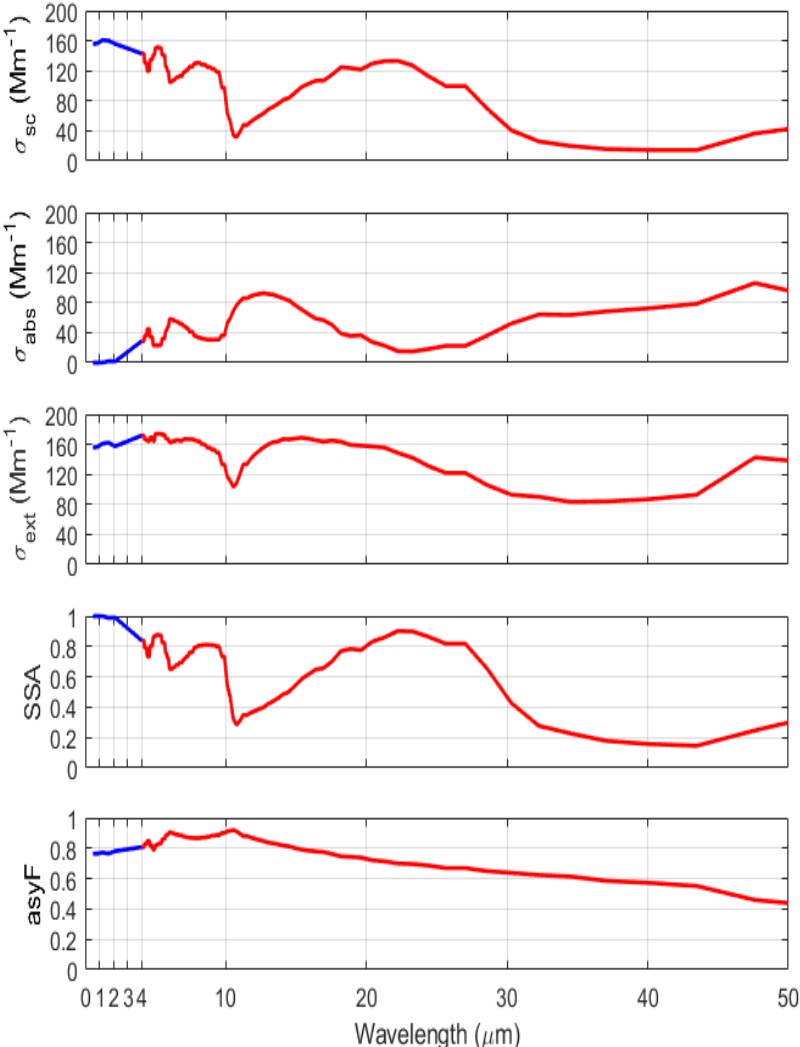

**Figure 2.** Spectral dependence of radiative properties obtained by the self-consistent scattering model for cirrus clouds: scattering, absorption,
extinction coefficients, single scattering albedo and asymmetry factor (from top to bottom) for a cirrus cloud of 08/12/2018 at 12 UTC,
measured in Barcelona. The colours indicate the shortwave range (blue; 0.2-4 $\mu$m) and longwave range (red; 4-50 $\mu$m).

Fig. 2 shows the spectral dependence of radiative properties obtained by the self-consistent scattering model for cirrus
clouds, which reflects the characteristic radiative properties of ice crystals. For example, it shows an absorption phenomenon
negligible in the spectral range between 0.2 and 2 $\mu$m and consequently, the single scattering albedo is approximately the



unity. As expected, the scattering phenomenon dominates the whole spectrum with respect to absorption except in the regions around 12 $\mu$m and for wavelengths higher than 30 $\mu$m (Yang et al., 2005, 2013). It is also noteworthy to mention that the single scattering albedo varies generally between 0.1 and 1 and at the working wavelength used in the model (0.532 $\mu$m) it has a value of 0.99 (Sun and Shine, 1994). This fact supports the hypothesis made previously that the absorption phenomenon is negligible.

The asymmetry factor presents much less variation as observed in the literature (Fu et al., 1998; Yang et al., 2005, 2013): it increases between 0.2 and 10 $\mu$m (in the range 0.75 - 0.95) and decreases afterwards (in the range 0.95 - 0.40).

### 3.2 The ARTDECO package

The Atmospheric Radiative Transfer Database for Earth and Climate Observation package (ARTDECO; https://www.icare.univ-lille.fr/artdeco/) is a numerical tool that gathers models and data for the 1D simulation of Earth atmosphere radiances and

fluxes from the ultraviolet to thermal infrared range (0.2-50 $\mu$m). It is developed and maintained at the Laboratoire d'Optique Atmosphérique (LOA) and distributed by the data and services center AERIS/ICARE (University of Lille), and funded by the TOSCA program of the French space agency (CNES). In ARTDECO, users can either access a library for the scene or use their own description through ASCII input files. Optical properties for aerosols and clouds can be computed. Then, the user can choose among available models to solve the radiative transfer equation and to compute radiative quantities corresponding

to the scene. ARTDECO is thus a flexible tool for remote sensing or radiative forcing applications, such as sensitivity studies, development and optimization of retrieval algorithms, evaluation of the future instruments performances, etc.

In this study, DISORT model is employed to solve the radiative transfer equation by discretising it (Stamnes et al., 2000). The ARTDECO environment allows us to solve the radiative transfer equation in two ways: 1) by introducing our own phase matrix

as a function of wavelength; 2) by using the Henyey-Greenstein function (Henyey and Greenstein, 1941), given extinction, single scattering albedo and asymmetry factor values over the whole spectral range in which the simulation will be done. Due to the lack of knowledge of the phase matrix of cirrus clouds with the observational measurements with which we work, the second option is chosen even though the Henyey-Greenstein function does not represent a good approximation to the real phase function, especially for ice crystals. Upward and downward radiative fluxes are calculated at different vertical Levels:

31 layers (0-20 km) in the shortwave (SW, 0.2-4 $\mu$m) and 40 layers (0-100 km) in the longwave (LW, 4-50 $\mu$m) spectra. These spectral/vertical ranges are adjustable, together with their spectral/vertical resolution. Cirrus forcings at the bottom-of-the-atmosphere (BOA) and top-of-the-atmosphere (TOA) have been calculated as:

$$BOA\,DRF = (F_c\downarrow -F_c\uparrow) - (F_o\downarrow -F_o\uparrow)\ at\ BOA \tag{4}$$

$$TOA\,DRF = (F_c\downarrow -F_c\uparrow) - (F_o\downarrow -F_o\uparrow) = -(F_c\uparrow -F_o\uparrow)\ at\ TOA \tag{5}$$

Where F$_c$ and F$_o$ are the radiative fluxes with and without the cirrus cloud, respectively. The ↓ and ↑ arrows indicate whether the fluxes are downward or upward, respectively. The simplification of Eq. 5 implies the assumption that the amount of the incoming solar radiation at the TOA is equal for both cases with and without aerosols. With this convention, a negative sign of





DRF implies a cirrus cooling effect independently of whether it occurs at the BOA or at the TOA. In this study, four types of
simulations are carried out: with gases only (G), with gases and aerosols in the layer closest to the surface (GA), with gases
and a cirrus cloud (GC) and finally, with everything (GAC). In Section 4.3, the radiative forcings of cirrus clouds in the full
atmosphere and in the whole spectral range considering aerosols (GAC-GA) or no aerosols (GC-G) are compared. Besides
aerosol optical properties, the radiative transfer model (RTM) DISORT is sensitive to atmospheric parameters such as the
relative humidity and the air temperature profiles, the surface emissivity and temperature or the aerosol vertical distribution
(Sicard et al., 2014).

The cirrus clouds are parameterized in the RTM model geometrically and optically, with the results obtained (Gil-Díaz et al.,
2024) and radiatively with the retrievals obtained with the self-consistent scattering model for cirrus clouds (see Section  3.1).
At the same time, the planetary boundary layer is characterized in the DISORT model geometrically and optically with the
MPLNET AER product and radiatively, on one hand, in the SW with the AERONET products and, on the other hand, in the
LW with the LITMS database.

### 3.2.1   Atmospheric profiles

The RTM DISORT model is run with atmospheric profiles (pressure, temperature, water vapor mixing ratio) obtained from
radiosondes launched in Barcelona at 00 and 12 UTC. The profiles of ozone concentration are obtained from Copernicus
Atmosphere Monitoring Service (CAMS) global reanalysis (EAC4) (Inness et al., 2019). EAC4 (ECMWF Atmospheric Com-
position Reanalysis 4) is the fourth generation ECMWF global reanalysis of atmospheric composition. Reanalysis combines
model data with observations from across the world into a globally complete and consistent dataset using a model of the
atmosphere based on the laws of physics and chemistry. The dataset is globally distributed with a horizontal resolution of
0.75°x0.75° and a vertical extension of 60 modes (from 1000 to 1 hPa). In exceptional cases, when no radiosondes or CAMS
data are available and for heights not covered by the radiosondes (generally above 30 km), the atmospheric profiles are taken
from the 1976 standard atmosphere (COESA et al., 1976).

### 3.2.2   Surface properties

In this study, a Lambertian surface is considered. On one hand, the corresponding surface albedo over the Barcelona region
is obtained for the SW range from AERONET and for the LW spectrum from CERES measurements. On the other hand, the
surface temperature is also taken from the SSF CERES product. In the parametrization of cirrus scenes, the surface albedo is
averaged seasonally and the surface temperature monthly, differentiating between values at daytime and nighttime, during the
period from 2018 to 2022. Surface albedo has been averaged seasonally due to its smoother temporal variation compared to
temperature, ensuring sufficient data availability.





### 3.2.3 Cloud/aerosol stratification

The vertical stratification of cloud/aerosols is reproduced according to the vertical profiles of MPLNET products. On one hand, the base and top of the cirrus clouds are obtained from (Gil-Díaz et al., 2024). On the other hand, the vertical distribution of the aerosols in the planetary boundary layer (PBL) is provided by MPLNET AER product. When this product is not available for that specific time period, it is assumed that aerosols are uniformly distributed throughout the aerosol layer, which extends up to 1.5 km, being the mean PBL height obtained in Barcelona over a 3-year period (Sicard et al., 2006).

## 3.3 The CLaMS-Ice Lagrangian microphysical cirrus model

CLaMS-Ice is a Lagrangian model for cirrus microphysics (Luebke et al., 2016; Baumgartner et al., 2022), intended to compute the evolution of ice microphysics along air parcel trajectories. The Chemical Lagrangian Model of the Stratosphere (CLaMS) (McKenna et al., 2002a, b; Konopka et al., 2004) performs the analysis of air mass back trajectories starting at arbitrary location in the atmosphere. The trajectories are derived from ECMWF windfields and are selectable over an arbitrary 280 time frame. Small scale temperature fluctuations not considered in the ECMWF wind fields are accounted for by superimposing temperature fluctuations according to (Podglajen et al., 2016). Next, the CLaMS-Ice model is run in the trajectories forward direction by using the two moment box-model developed by (Spichtinger and Gierens, 2009) to simulate cirrus cloud development. The two-moment scheme includes homogeneous as well as heterogeneous nucleation of ice, depositional growth of ice crystals, their evaporation, aggregation, and sedimentation (sedimentation parametrization after (Spichtinger and Cziczo, 285 2010); the sedimentation parameter is set to 0.97). Heterogeneous freezing starts at a critical supersaturation of 120%. The initial concentration of ice nucleating particles is prescribed in the model (mean value: 0.01 cm-3). As ice particles evaporate, ice nucleating particles are released back into the air parcel. The model predicts the ice number concentration and the ice water content. If ice is already present in the ECMWF data at the start of the trajectory, CLaMS-Ice treats this as pre-existing ice. Two-moment scheme only considers the trajectories that end at T < 238K. If a part of the trajectory existed at T > 238 K 290 before crossing into the colder cirrus environment, then the forward model is initialized with pre-existing ice from mixed phase clouds, if present in the IWCs found in the ECMWF data. CLaMS-Ice proceeds with pre-exixsting ice as with newly formed cirrus clouds.

## 4 Five years of cirrus retrievals

This study analyses the radiative properties of cirrus clouds that were previously characterized geometrically, thermally and 295 optically in (Gil-Díaz et al., 2024). In the latter paper, 203 cases were analyzed. Here, atmospheric scenes with only one cirrus cloud in the vertical profile are studied. 125 single-cirrus cloud scenes are found, that constitute 61% of all cases. Case selection involves discarding cirrus scenes where a mid-level cloud has been detected below the cirrus cloud. Mid-level cloud detection was performed through visual analysis during the application of the two-way transmittance method (Gil-Díaz et al., 2024). This selection ensures the simulated cirrus scenes accurately represent reality (see Section 4.2).





## 4.1 Cirrus radiative properties

After having carried out the identification of 125 high-altitude cirrus scenes with only a cirrus cloud in the vertical profile (being the 39% of cirrus clouds measured at daytime), the self-consistent scattering model for cirrus clouds is applied to obtain their radiative properties (see Section 3.1). In this section the radiative properties of cirrus clouds are presented and discussed. For this purpose, probability distributions of the following averages of radiative properties for each cirrus scene: ice water content, single scattering albedo and asymmetry factor, calculated at 0.55 $\mu$m are shown in Fig. 3. For each case, the IWC and the radiative properties of the cirrus clouds have been obtained using the mean cloud temperature and the effective extinction coefficient of the column for each model vertical layer in which the cloud is extended. This calculation is performed by averaging the vertical profile of the cloud extinction coefficient over half the cloud thickness and centred on the maximum peak to obtain a value representative of the entire cirrus.

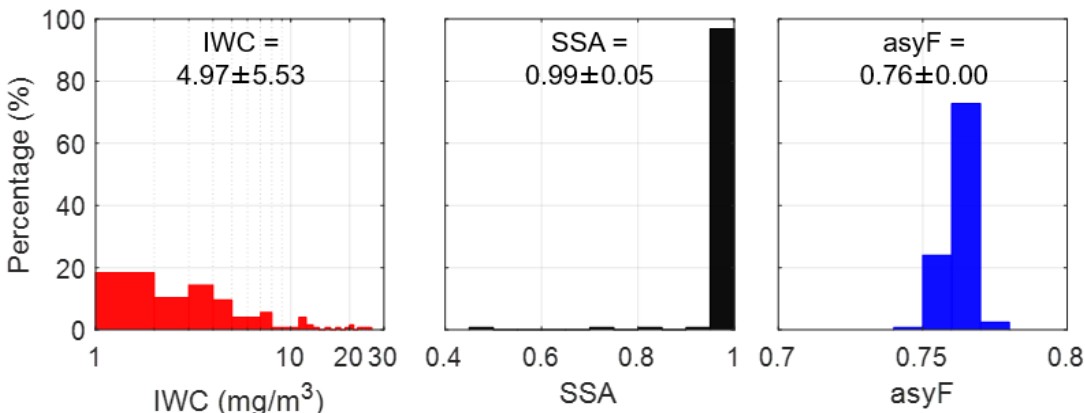

**Figure 3.** Probability distribution of averages of (left) ice water content (IWC), (center) single scattering albedo (SSA) and (right) asymmetry factor (asyF) for each cirrus scenes at 0.55 $\mu$m, for cirrus clouds measured from 2018 to 2022 in Barcelona. The X-Axis of the left histogram in in logarithmic scale.

In Fig. 3 one observes that cirrus clouds have an IWC between 0.03 and 30 mg/m$^3$, being characteristic of mid-latitude cirrus clouds (Field et al., 2005, 2006; Baran et al, 2011b; Sourdeval, 2012; Kramer et al., 2016, 2020). Where the average of IWC is ~5 mg/m$^3$, being a value close to 3 mg/m$^3$, which is the central value of the mid-latitude ice cloud distributions obtained by (Sourdeval, 2012) and the mean value of IWC for temperatures between 210 and 235K found in (Kramer et al., 2016). The single scattering albedo of most cirrus clouds (97%) has a value between 0.95 and 1, as expected at 0.55 $\mu$m (Hess and Wiegner, 1994; Sun and Shine, 1994; Yang et al., 2013; Hemmer, 2018). Although there are 3 cases of cirrus clouds whose SSA < 0.9. These cases correspond to sub-visible cirrus clouds with an IWC less than 1 mg/m$^3$. These 3 cases have in common that the cirrus cloud extends in two vertical layers of the model and in one of the layers, the resulting value of the effective column extinction coefficient is less than 1 Mm$^{-1}$. In this layer, a low value of SSA is obtained, associated to its low





value of the extinction coefficient and consequently, when averaging the radiative properties in the two layers for each cirrus

scene, the values of SSA < 0.9 observed in Fig. 3 are obtained. Therefore, the ensemble scattering model for cirrus clouds might associate the low effective column extinction values to super-cooled liquid water content in the cirrus clouds. The asymmetry factor of cirrus clouds varies between 0.7 and 0.8, with an average of 0.76, as expected at 0.55 $\mu$m (Hess and Wiegner, 1994; Sun and Shine, 1994; Yang et al., 2013; Hemmer, 2018).

## 4.2 Validation of radiative fluxes

The validation of the ARTDECO package is performed by comparing the simulated radiative fluxes with observed ones. For that purpose, the radiative fluxes from ARTDECO were recalculated in the range 0.305-2.8 $\mu$m corresponding to the spectral range of the SolRad-Net pyranometer (BOA), and in the range 5-200 $\mu$m corresponding to the spectral range of CERES (TOA). The scatter plot of the simulated vs. observed SW downward radiative fluxes of cirrus clouds classified according to their cloud optical depth (Sassen and Cho, 1992), at the surface and the LW upward radiative fluxes at TOA are shown in Fig. 4.

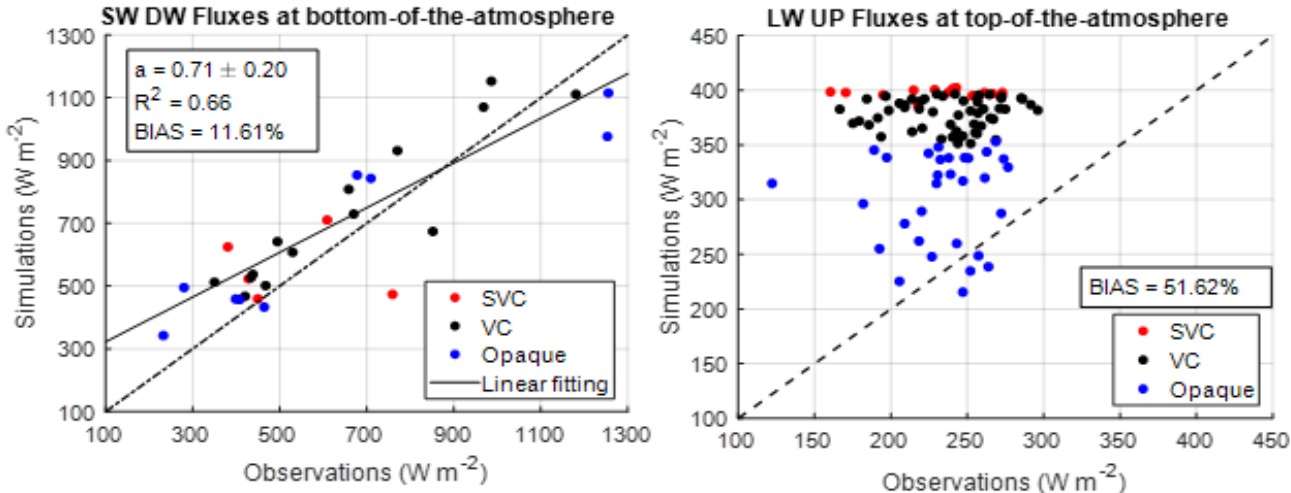

**Figure 4.** Comparison of (left) simulated shortwave downward (SW DW) radiative fluxes at the bottom-of-the-atmosphere and SolRad-Net observations; (right) simulated longwave upward (LW UP) radiative fluxes at top-of-the-atmosphere and CERES observations. The black dashed line is the curve with the slope unity and the black solid line is the linear fitting of the fluxes (y=ax+b, being a the slope and $R^2$ its determination coefficient). The cirrus clouds have been classified according to (Sassen and Cho, 1992) criteria: sub-visible (SVC; COD < 0.03), visible (VC; 0.03 < COD < 0.3) and opaque clouds (Opaque; COD > 0.3).

The validation of the SW downward radiative fluxes is performed with 59% of the cirrus clouds measured at daytime and the validation of the LW upward radiative fluxes with 81% of all cirrus clouds considered in this study. The cases of cirrus clouds that could not be validated are due to the lack of observations. In addition, to reduce the effect of cloud movement on the radiation measurement with the pyranometer, the observed radiation fluxes are averaged over 30 minutes. In Fig. 4 (left)




it can be seen that most downward radiative fluxes calculated with the DISORT model overestimate the SolRad-Net obser-
vations: the mean and standard deviation of the simulated fluxes are $694 \pm 247$ Wm$^{-2}$, while it is $621 \pm 283$ Wm$^{-2}$ for
the observations. This translates into a systematic BIAS of +11.61% with a steep slope of the linear regression (a = 0.71 $\pm$
0.20). The overestimation may be related to the error associated with variables obtained by the self-consistent scattering model
for cirrus clouds, as cirrus clouds govern the radiation interactions in these simulations, because of their cloud optical depth.
The ensemble scattering model for cirrus clouds has a large error for small ice crystals (less than 100 $\mu$m), corresponding to
cirrus clouds with low IWC values (Liou et al., 2008). In particular, the model tends to underestimate the IWC for mid-latitude
cirrus clouds (Baran and Labonnote, 2007). Therefore, when the IWC is lower, the extinction of cirrus clouds is smaller as
demonstrated in (Fu, 1996; Heymsfield et al., 2014) and, consequently, allows more radiation to pass into the atmosphere than
actually does.

As for validation in the longwave spectrum, CERES measurements have been selected based on their geographical position.
Specifically, the measurements closest to the Barcelona lidar station have been selected, despite the fact that the hour of
the measurement does not correspond to the exact hour of the atmospheric scene. The hourly difference between simulation
and observation is not relevant in the validation of the LW radiation fluxes, since it is almost constant during daytime hours
(Sicard et al., 2014). In Fig. 4 (right) a large horizontal dispersion can be observed. In addition, a general overestimation of
the CERES observations with the ARTDECO simulations is produced, being for simulated fluxes $357 \pm 46$ Wm$^{-2}$ and for
the observations $235 \pm 32$ Wm$^{-2}$. In our case, the large BIAS = +51.62 % obtained could be due to the spatial resolution
of the observed measurements taken, which lie on a 0.2x0.2º grid around the Barcelona station and may cover part of the
Mediterranean Sea. In addition, as shown in (Gil-Díaz et al., 2024) most of the cirrus clouds are visible and therefore their
horizontal expansion is smaller than the cirrus clouds that form at higher altitudes (well-known as sub-visible cirrus clouds)
(Kramer et al., 2020), which makes them more difficult to see from top-of-the-atmosphere. Hence, the comparison of simulated
radiative fluxes and CERES observations is not as trivial and conclusive as with SolRad-Net observations, since the CERES
satellite can observe a slightly different atmospheric situation, as mentioned above. Not to mention the limitations of the 1-D
radiative transfer model DISORT to represent an irregular composition of broken and/or overlapping clouds that the CERES
satellite could observe.

### 4.3 Study of the influence of aerosols on radiative simulations of cirrus clouds

The cloud radiative forcing is often calculated as the difference between radiative fluxes under cloudy and cloudy-free condi-
tions, without considering the aerosols found in the layer of the atmosphere closest to the surface (Ramanathan et al., 1989;
Hartmann et al., 2001; Barja and Antuña, 2007; Yang et al., 2007; Lee et al., 2009; Campbell et al., 2016; Lolli et al., 2017).
In this way, simpler simulations are carried out, as it is only necessary to characterize the cloud. It is known that radiation
does not interfere linearly with the components of the atmosphere: clouds, aerosols and gases. Therefore, in this subsection we
analyse whether or not the insertion of aerosols in the lowest atmospheric layer in cirrus cloud scenes modifies the calculation





of cirrus radiative forcings. For this purpose, the net forcing (NET; SW+LW) in the atmosphere (ATM; TOA-BOA) of cirrus clouds calculated with simulations in which aerosols have been considered or not is compared, as shown in Fig. 5.

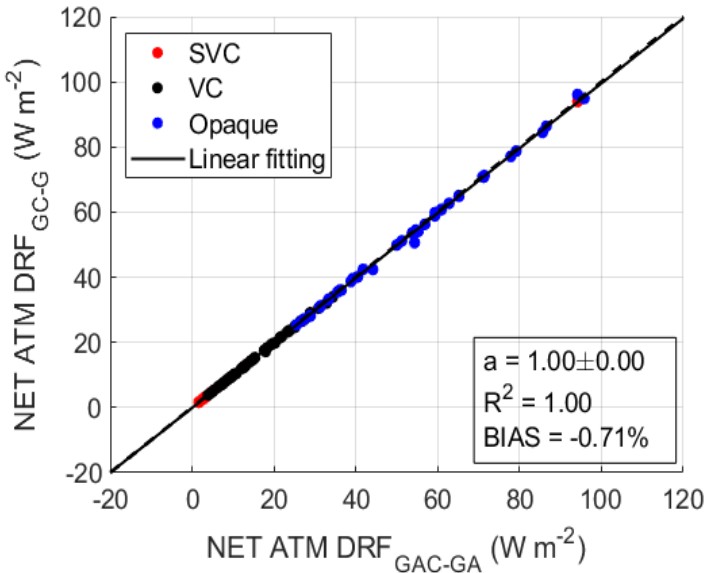

**Figure 5.** Comparison of net radiative forcings in the full atmosphere (NET ATM DRF) between simulations made with (X-axis) and without (Y-axis) aerosols. The black dashed line is the curve with the slope unity and the black solid line is the linear fitting of the radiative forcings (y = ax+b, being a the slope). The cirrus clouds have been classified according to (Sassen and Cho, 1992) criteria: sub-visible (SVC; COD < 0.03), visible (VC; 0.03 < COD < 0.3) and opaque clouds (Opaque; COD > 0.3).

Fig. 5 shows that the NET ATM radiative forcings calculated with and without aerosols fit well, with the most of the points
lying slightly above on the curve with the slope unity. As a consequence, its linear fitting rounding to the tenth has also a unity slope, with a $R^2$ value of 1.00. In addition, the mean and standard deviations of simulations reflect that there is an negligible underestimation of the forcings when not considering aerosols, with values for the simulation with aerosols (X-axis) of $+27.21 \pm 23.94$ Wm$^{-2}$ and without aerosols (Y-axis) of $+27.02 \pm 23.86$ Wm$^{-2}$. Furthermore, the BIAS is -0.71%, being a low value, possibly due to the distance between the aerosol layer and the cirrus cloud (being on average $6.76 \pm 2.24$ km). With these
results where the aerosol layer was well distinguished vertically from the cirrus clouds, the simplification of the atmospheric scenes can be made without considering aerosols, but to be more rigorous, in the following results, only the forcings in which aerosols are present will be considered. In the other case where the aerosols are vertically closer to the clouds (lower than 1 km, being the minimum distance found between the cirrus cloud and the aerosol layer), the simplification of not considering aerosols in the calculation of cloud forcings may not be valid, leading to a significant underestimation of cloud forcings.





## 4.4 Radiative forcings of cirrus clouds depending on COD

In this section, only radiative forcings of cirrus clouds calculated with simulations in which aerosols are present, $DRF_{GAC-GA}$, will be considered and will be denoted as DRF. Special attention will be paid to net radiative forcings of cirrus clouds at daytime because they are the only clouds that can readily cool or warm the top and bottom of atmosphere, during daytime, depending

on their properties (Campbell et al., 2016). In order to quantify this phenomenon, cirrus clouds at daytime and nighttime have been distinguished. The net radiative forcings of cirrus clouds at nighttime and daytime, at BOA and TOA are shown in Fig. 6.

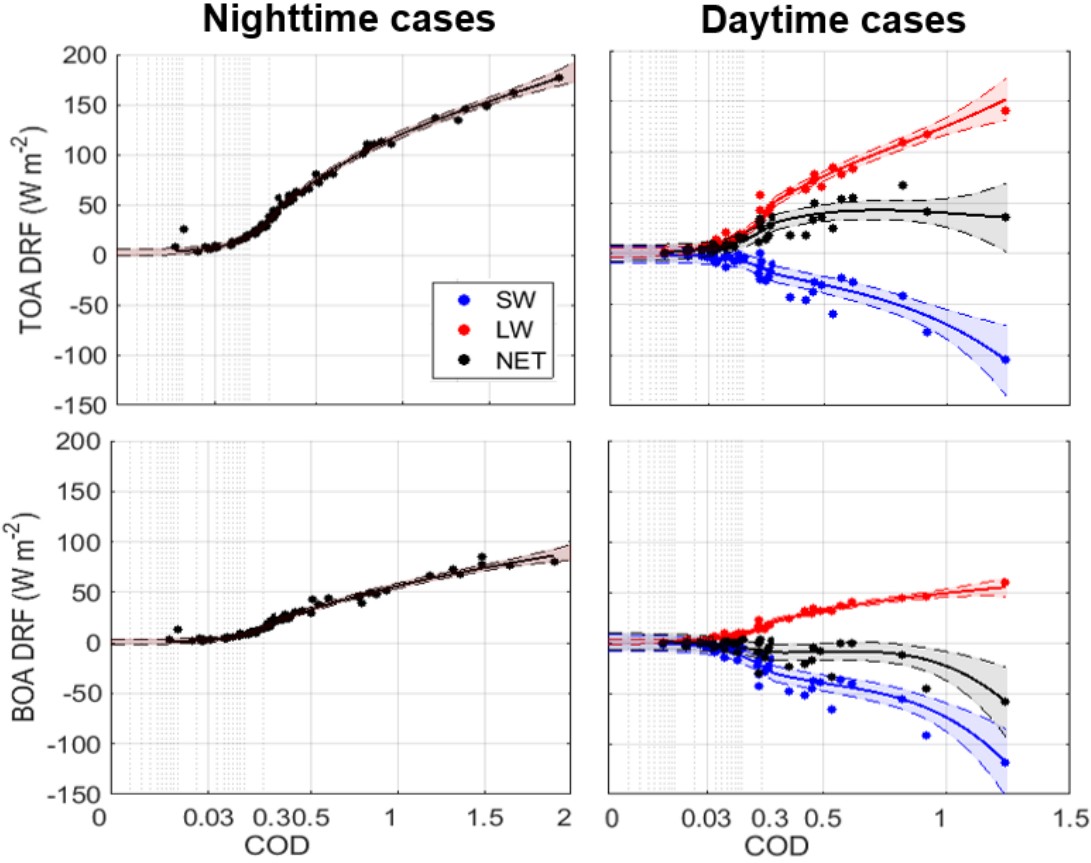

**Figure 6.** Distribution of radiative forcings of cirrus clouds at (left) nighttime and (right) daytime, at (bottom) bottom-of-the-atmosphere (BOA) and (top) top-of-the-atmosphere (TOA), in function of their cloud optical depth (COD). Shortwave (SW), longwave (LW) and net (NET = SW + LW) components of radiative forcings have been distinguished. For COD < 0.3 a logarithmic scale has been considered in order to discern more clearly sub-visible and visible cirrus clouds (Sassen and Cho, 1992). The solid line corresponds to the polynomial fitting performed on the data set. The shaded area represents the region with a 95% probability of containing the points, adjusted by the mean absolute value of the differences between the actual and fitted values.



In Fig. 6 one observes a positive trend between the net radiative forcings with the COD, where the thicker cirrus clouds contribute more to the overall forcing budget, as has been observed in other studies (Barja and Antuña, 2007; Lee et al., 2009; Campbell et al., 2016; Lolli et al., 2017). Some COD gaps are also found, because the cirrus observations considered do not
have a homogeneous and equidistant distribution of COD. At nighttime, the net cirrus forcing at TOA is approximately twice that at BOA, being always positive as expected. Cirrus clouds at nighttime act as a cover in the atmosphere, they do not let through all the infrared radiation emitted by the Earth as it cools, inducing a warming of the atmosphere. This warming in function of COD is faster at TOA than at BOA because the atmosphere at BOA is strongly influenced by the surface, which acts as a black body emitting infrared radiation at nighttime. Consequently, the heating at BOA is milder than at TOA.


At daytime, at TOA, the net radiative forcing remains positive for all cirrus clouds, dominating the positive longwave component. In contrast, at BOA, the net radiative forcing is almost always negative (only 20% of the cases show a positive net forcing, whose value is close to 0), being the outgoing shortwave radiation in the presence of cirrus clouds larger than in cirrus cloud free conditions. The albedo effect overcomes the greenhouse effects in the SW range because of low absorption capacity
of the small crystals, as shown in Fig. 2. These changes of sign which occur mostly for thin cirrus clouds (67% of the cases) have already been observed in other studies such as (Campbell et al., 2016; Lolli et al., 2017; Kramer et al., 2020), where the dominant factor in the change of sign of forcing is unclear. Multiple factors are involved, from the optical and radiative properties of the cirrus, to the solar zenith angle or the surface temperature and surface albedo (Wolf et al., 2023).

In order to complete this analysis, the net radiative forcings in the full atmosphere are analysed for cirrus clouds at nighttime and daytime, as shown in Fig. 7.





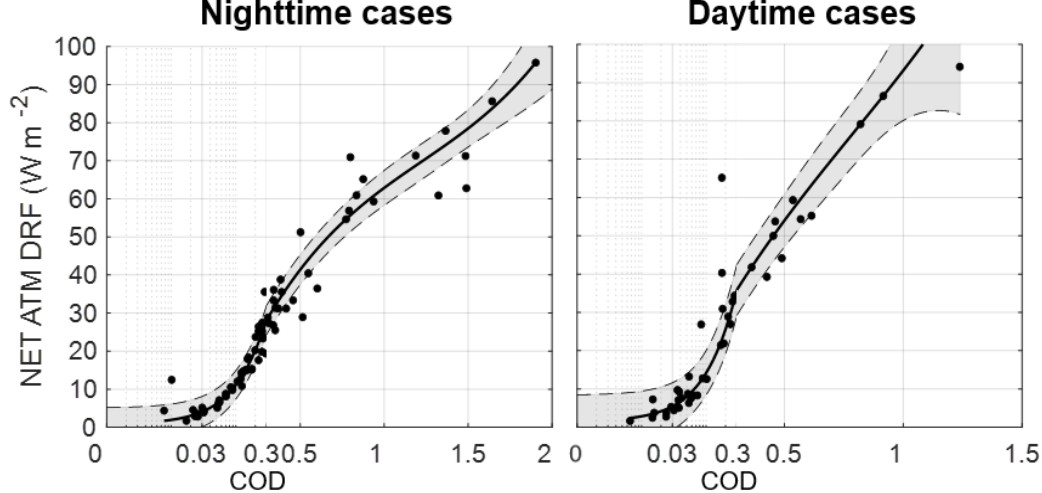

**Figure 7.** Distribution of net radiative forcings of cirrus clouds in the full atmosphere at (left) nighttime and (right) daytime, in function of their cloud optical depth (COD). For COD < 0.3 a logarithmic scale has been considered in order to discern more clearly sub-visible and visible cirrus clouds (Sassen and Cho, 1992). The solid line corresponds to the polynomial fitting performed on the data set. The shaded area represents the region with a 95% probability of containing the points, adjusted by the mean absolute value of the differences between the real and fitted values.

Fig. 7 shows a net warming of the atmosphere (always positive forcing) for cirrus clouds at nighttime and daytime, being radiation escape lower in the presence of cirrus clouds in the full atmosphere. This phenomenon could have been perceived in the previous figure (Fig. 6) as the forcing at TOA was always higher than at BOA. It also results that the atmosphere warms faster in function of the COD during the daytime than at nighttime (see their regression slopes), as expected due to the contribution of solar radiation to the net forcing. The net forcing in the full atmosphere fits very well with the polynomial regressions for both time periods, being at nighttime $R^2 = 0.99$ and at daytime $R^2 = 0.97$, although some instances outside the shaded area are observed. These strong variations of the radiative forcing for cirrus clouds with very similar COD are due to the consideration in the simulations of different radiative properties of the cirrus clouds, thermodynamic profiles, surface temperature and surface albedo values for each cirrus scene.

Then, the radiative forcings for cirrus clouds at nighttime and daytime, which are classified according to (Sassen and Cho, 1992) criteria, are quantified, as shown in Table 1.



| Type | Nighttime | | | | Daytime | | | |
|---|---|---|---|---|---|---|---|---|
| | COD | BOA | TOA | ATM | COD | BOA | TOA | ATM |
| Sub-visible | 0.02±0.01 | 4.13±3.99 | 8.57±7.35 | 4.44±3.37 | 0.02±0.01 | -0.61±1.21 | 3.24±1.40 | 3.85±1.90 |
| Visible | 0.15±0.08 | 11.77±6.03 | 27.30±13.29 | 15.53±7.74 | 0.11±0.08 | -3.89±7.17 | 14.33±10.62 | 18.22±14.67 |
| Opaque | 0.76±0.47 | 43.74±20.56 | 93.05±38.06 | 49.31±20.02 | 0.75±0.39 | -20.38±18.82 | 39.42±15.78 | 59.80±18.56 |
| Total | 0.37±0.43 | 23.02±21.23 | 50.06±42.99 | 27.04±22.39 | 0.28±0.37 | -8.57±14.95 | 18.94±16.95 | 27.51±26.63 |

**Table 1.** Average and standard deviation of radiative forcings of cirrus clouds (Wm$^{-2}$) at bottom-of-the-atmosphere (BOA), top-of-the-atmosphere (TOA) and in the full atmosphere (ATM), at nighttime and daytime, classified with (Sassen and Cho, 1992) criteria in Barcelona. Cloud optical depth values are obtained from (Gil-Díaz et al., 2024).

In Table 1 it is discernible that thicker cirrus clouds produce a higher forcing than thinner clouds, as observed above. At nighttime, cirrus clouds produce an average net warming in the full atmosphere of +27.04±22.39 Wm$^{-2}$, with opaque cirrus clouds being the main source. At daytime, cirrus clouds generally cool the BOA and warm the TOA, resulting in a warming of the full atmosphere. The radiative forcing at BOA ranges between -58 and +4 Wm$^{-2}$ for all cirrus. In particular, for thin cirrus clouds the radiative forcing is in a range from -30 to +3 Wm$^{-2}$, being a similar range to (Lee et al., 2009), covering from -20 to 0 Wm$^{-2}$. Therefore, shortwave negative forcing generally dominates at the BOA, with an average of -3.18±6.48 Wm$^{-2}$ for thin cirrus clouds, being slightly lower than -1.35 Wm$^{-2}$ (Lee et al., 2009). The radiative forcing at TOA ranges between +1 and +67 Wm$^{-2}$ for all cirrus, being a wider interval than (Campbell et al., 2016; Lolli et al., 2017; Kramer et al., 2020; Kienast-Sjögren et al., 2016). In particular, for thin cirrus clouds the radiative forcing is in a range from +1 to +36 Wm$^{-2}$, being the maximum value considerably higher than the value of +5.71 Wm$^{-2}$ (Campbell et al., 2016) or +10 Wm$^{-2}$ (Kramer et al., 2020). The average of the radiative forcing for thin cirrus clouds is +11.90±10.45 Wm$^{-2}$, being a close value compared to them obtained in (Kienast-Sjögren et al., 2016), that cover in average from 6.2 to 11 Wm$^{-2}$, although they are also significantly higher than the value of +1 Wm$^{-2}$ (Lee et al., 2009). Despite the differences found, the values are in agreement in magnitude with other studies such as (Ackerman et al., 1988; Jensen et al., 1994; Lee et al., 2009; Berry and Mace, 2014; Campbell et al., 2016; Kienast-Sjögren et al., 2016; Kramer et al., 2020). The average net warming in the full atmosphere is a little higher at daytime, with an average value of +27.51±26.63 Wm$^{-2}$. This difference is apparently not related with the fraction of opaque cirrus, as the percentage of opaque cirrus is lower during nighttime (28%) than at daytime (38%).

### 4.5 Radiative forcings of cirrus clouds depending on solar zenith angle

In this section, only net radiative forcings of cirrus clouds at daytime, will be considered. The distinction between the shortwave and longwave spectral ranges will not be made because radiative forcing does not depend on the solar zenith angle in the longwave spectrum (Lee et al., 2009; Wolf et al., 2023). The net radiative forcing of cirrus clouds during daytime at the BOA and TOA, together with the results of a brief sensitivity study, in which all parameters of the simulations except the solar zenith angle are kept constant, are shown in Fig. 8.





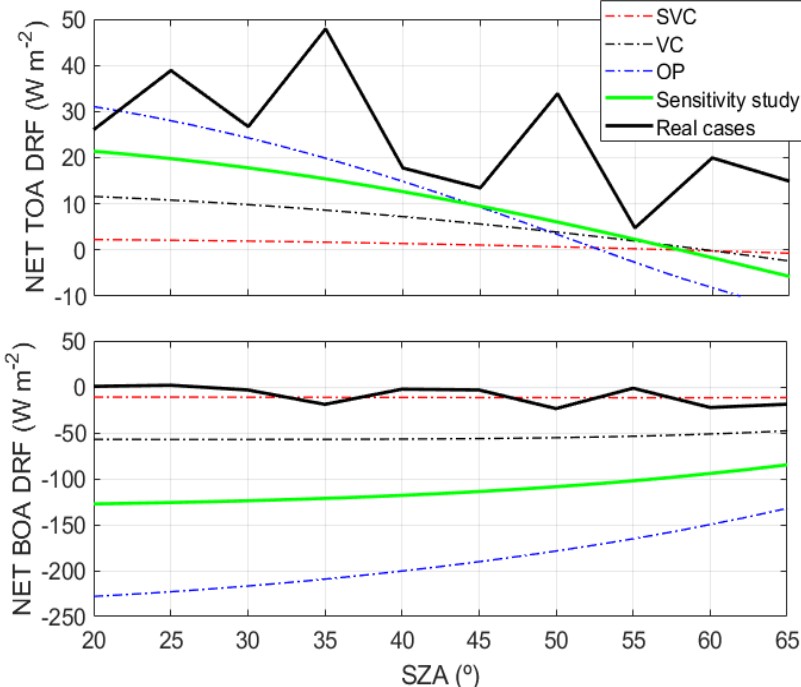

**Figure 8.** Distribution of the net radiative forcings of cirrus clouds at daytime at (top) top-of-the-atmosphere (TOA) and (bottom) bottom-of-the-atmosphere (BOA) in function of their solar zenith angle (SZA) resulting from the (dashed curves correspond to for each cirrus cloud type according to the (Sassen and Cho, 1992) criteria and green curve correspond to the mean values for all daytime cirrus clouds) sensitivity study and (black curve) the real cases.

In the sensitivity study, three cloud types have been considered, according to the (Sassen and Cho, 1992) criteria, where the cloud optical depth is shown in Table 1 for the daytime cirrus clouds. The geometrical, thermal and optical properties correspond to the mean values of the cirrus clouds measured at Barcelona lidar station (Gil-Díaz et al., 2024) and the radiative

properties are obtained from the mean values resulting from the statistic. The longwave surface albedo considered is 0.0157 and the surface temperature is 28ºC. The thermodynamic profiles have been selected from the 1976 standard atmosphere (CO-ESA et al., 1976). By keeping constant all other properties of cirrus clouds scenes, the results are expected to be exclusively due to the variation in the value of the solar zenith angle. On the other hand, in the real cases that have been analysed in Section 4.4 and Section 4.5 as a function of cloud optical depth, the net radiative forcing values for the three cloud types have

been averaged, with a SZA resolution of 5º.

In Fig. 8 one observes a higher variability of net radiative forcing for the real cases than for the sensitivity study. This variability could be explained by other parameters that are considered in the simulations, such as cloud optical depth, cirrus radiative properties, surface albedo and temperature (Sicard et al., 2014; Wolf et al., 2023). In addition, a generally higher



mean net radiative forcing is discerned for the real cases than for the sensitivity study results, especially at BOA despite having taken the average properties resulting from the statistics. At TOA, a slight downward tendency of the net radiative forcing is obtained as the SZA increases, as found in (Wolf et al., 2023). As SZA increases, cloud solar extinction is enhanced regarding thermal effects (Campbell et al., 2016). All mean net radiative forcing values for the real cases are positive, but there is a large fluctuation in certain values of SZA. Moreover, there is no crossover where the mean net radiative forcing shifts from positive

to negative values between SZA of 20 to 65º. On contrary, for the results from the sensitivity study, a change of sign is observed at 58º, fixing well with results from (Campbell et al., 2016; Lolli et al., 2017). At BOA, most net radiative forcing values are negative, presenting a slight increasing trend unlike the TOA. Since the angle of incidence of the incoming solar radiation increases, the incident solar radiation is lower and the scattering produced by the ice crystals increases because optical path is larger (Lee et al., 2009; Wolf et al., 2023). This enhancement of the extinction of incident solar radiation, which is lower

results in a reduction of the net warming effect at TOA and the net cooling effect of the atmosphere at BOA. The change from positive to negative values of the mean net radiative forcing is only observed for the real cases, being SZA crossover of 25º, a considerably lower value than that observed for the results of the TOA sensitivity study.

### 4.6   Case study of radiative forcings of an evolving cirrus cloud

In this section the role of time is added to the analysis and the radiative forcing produced by a cirrus cloud is studied along its

back-trajectory. The final objective of this case is to simultaneously analyse the evolution of different physical agents such as the surface albedo, the solar zenith angle, the cloud optical depth or the ice water content in the quantification of the cirrus cloud radiative forcing. The case of study corresponds to the back-trajectory of a cirrus cloud measured at Barcelona lidar station on 11/02/2019 at 02:03 UTC (Gil-Díaz et al., 2024), where simultaneous measurements of the MPL and CALIPSO were performed. To simulate the evolution of the cirrus cloud as realistic as possible, its microphysical properties along its back-

trajectory are calculated with the CLaMS-Ice model. This model provides apart from the basic back-trajectory variables such as temperature, altitude, geographic coordinates and time, microphysical properties like ice water content, ice crystal number concentration or ice nuclei concentration. Considering the temperature provided by the model as the mid-cloud temperature and together with the ice water content, the radiative properties of cirrus clouds are specifically calculated with Eq. 1 and Eq. 2. Assuming the cirrus cloud geometric thickness decreases linearly, using the CALIPSO measurements of 1 km on 10/02/2019

14 UTC and 1.74 km on 11/02/2019 2 UTC, and considering that the geometrical thickness is 0 km when the IWC is null, the COD is estimated as the product of the extinction coefficient and the geometric thickness. The back-trajectory of the cirrus cloud and the properties: ice water content, mid-cloud temperature, mid-cloud altitude, cloud geometrical thickness and cloud optical depth are shown in Fig. 9.





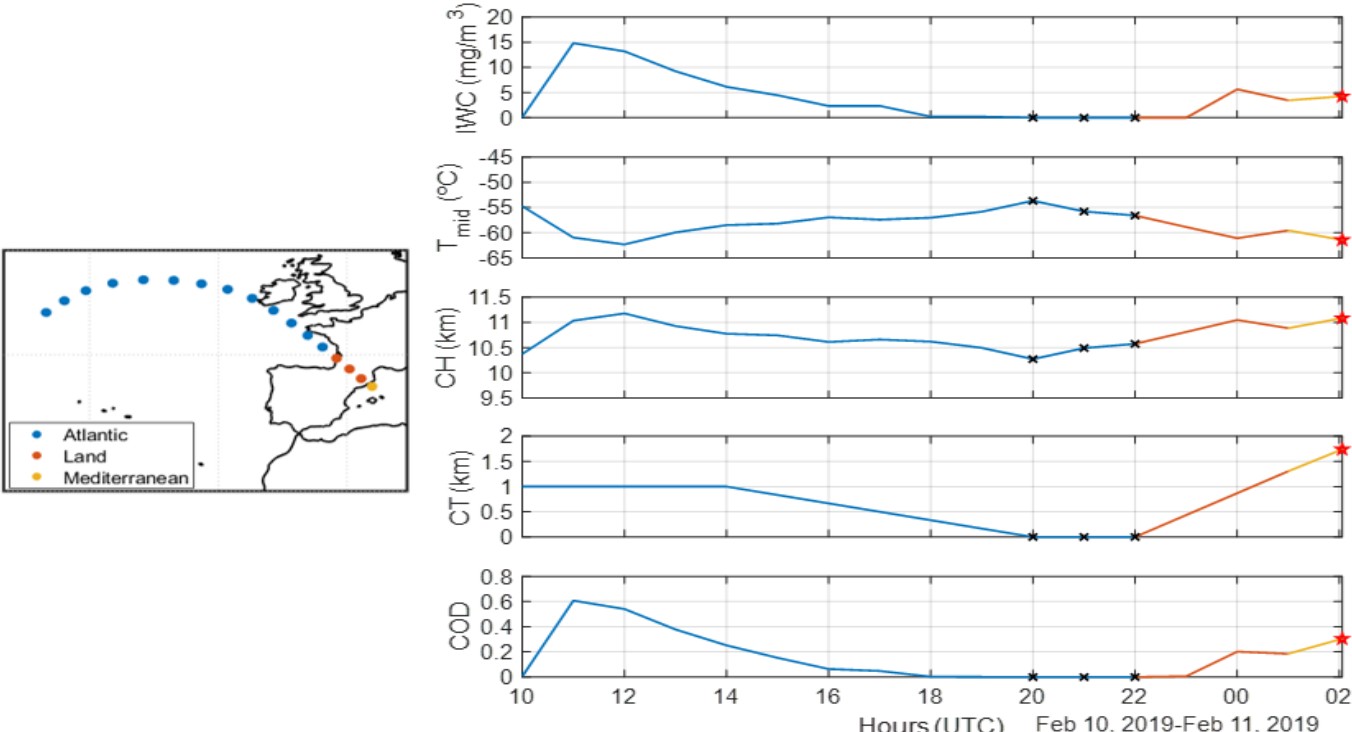

**Figure 9.** (Left) Hourly back-trajectory of the cirrus cloud for the last 16 hours before arrival in Barcelona. (Right) Evolution of cirrus cloud properties along its back-trajectory: ice water content (IWC), mid-cloud temperature ($T_{mid}$), mid-cloud altitude (CH), cloud geometrical thickness (CT) and cloud optical depth (COD) (from top to bottom). Colours indicate the type of surface where the cloud is over, black crosses mark the non-existence of cirrus cloud during those hours and the red star points to the case measured by the MPL and CALIPSO in (Gil-Díaz et al., 2024).

In Fig. 9 one observes that the cirrus cloud comes from the Atlantic Ocean, passing through part of France and Barcelona to reach the Mediterranean Sea. The selected points of the trajectory are spaced 1 hour apart backward from 11 February 2019 at 02 UTC. During this journey, the air mass which corresponds to the cloud undergoes a rise in height, reaching a minimum in temperature and a COD of 0.6. After this initial cirrus cloud fast growth, the cloud gradually fades away until 20 UTC, where the CLaMS-Ice model gives a null ice water content. Afterwards, a new cirrus begins to form over land surface and to grow until 00 UTC and remains relatively stable.

Taking advantage of the overlap between the back-trajectory of this cirrus cloud and the CALIPSO satellite, a brief evaluation of the ice water content is carried out on 11/02/2019 02:03:20 UTC. The difference in IWC between the CLaMS-Ice output and the CALIPSO measurement is 0.88 mg/m$^3$, resulting in a relative deviation of 29% for the CLaMS-Ice output. This difference obtained by CLaMS-Ice and CALIPSO, well within a factor of two, is considered reasonable given the different parameterizations involved in the calculation of the IWC and their respective uncertainties. To fully characterize the cirrus





cloud scenes along its back-trajectory, it is considered that the height from the CLaMS-Ice model is the mid-cloud height and the surface albedo is constant over each of the three distinct surface types. The albedos of the Atlantic Ocean and the Mediterranean Sea are estimated with data from the CERES satellite in the SW and the land albedo is assumed to be that of Barcelona. With all these assumptions, radiative simulations are calculated with the ARTDECO package and the radiative

fluxes at TOA, BOA and the full atmosphere are shown in Fig. 10.

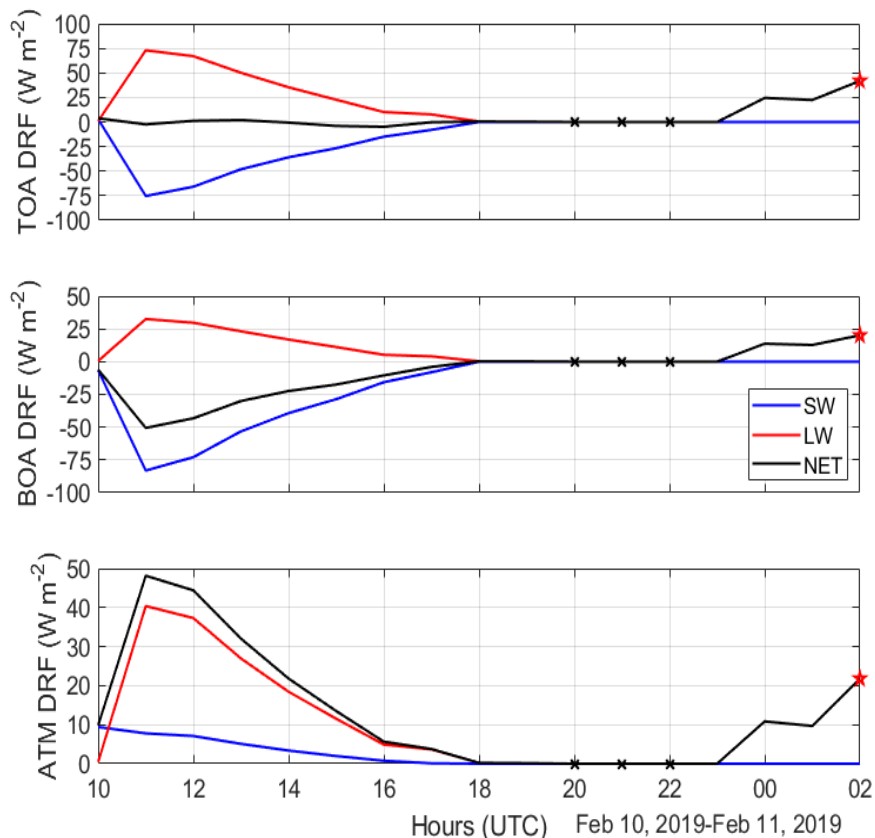

**Figure 10.** Distribution of radiative forcings of the cirrus cloud along its back-trajectory at (top) top-of-the-atmosphere (TOA), (centered) bottom-of-the-atmosphere (BOA) and (bottom) in the full atmosphere (ATM). Shortwave (SW), longwave (LW) and net (NET = SW + LW) components of radiative forcings have been distinguished. Black crosses mark the non-existence of cirrus cloud during those hours.

Fig. 10 shows that at TOA the net radiative forcing is close to zero with values shifting between positive and negative during the first hours where the cirrus cloud is over the Atlantic Ocean. During this period, the SW component is almost completely balanced by the LW component and it is zero when there is no incident solar radiation, i.e. at nighttime, therefore the net forcing corresponds to the LW component. At nighttime, the net direct radiative forcing at TOA is also approximately double

than at BOA, in agreement with previous results. At BOA, the net forcing changes from negative to positive, since the incident



solar radiation produces that the albedo effect overcomes the greenhouse effect during daytime. In the full atmosphere, the radiative forcings are always positive, as the forcing at TOA is higher than at BOA. In summary, these simulations reveals the evolution of the net radiative forcing produced by the cirrus cloud, going at TOA from values close to 0 to +42 Wm$^{-2}$, at BOA from negative values, whose minimum is -51 Wm$^{-2}$ to positive values reaching a maximum of +20 Wm$^{-2}$ and in the full

atmosphere varying between values close to 0 to +48 Wm$^{-2}$, being the maximum. The complexity of calculating the radiative forcing of a cirrus cloud lies in the fact that this value is highly sensitive to its scene cloud properties like cloud optical depth, solar zenith angle or surface albedo as seen in this case study.

Finally, we compare the radiative forcing of the cirrus cloud (red star mark) measured by MPL at the Barcelona lidar station,

by CALIPSO satellite at 78 km from Barcelona lidar station (Gil-Díaz et al., 2024) and with the properties obtained with CLaMS-Ice, as shown in Table 2.

| Database | Properties | | | | NET DRF (Wm$^{-2}$) | | |
|---|---|---|---|---|---|---|---|
| | $T_{mid}$ (ºC) | $\sigma_{ext}$ (Km$^{-1}$) | IWC (mg/m$^3$) | COD | BOA | TOA | ATM |
| MPL | -60.71 | 0.17 | 4.11[a] | 0.26 | 18.68 | 47.72 | 29.04 |
| CALIPSO | -63.73 | 0.16 | 22.15 | 0.23 | 17.43 | 44.17 | 26.74 |
| CLaMS-Ice | -61.44 | [b] | 4.26 | 0.30 | 21.49 | 55.34 | 33.86 |

**Table 2.** Properties (mid-cloud temperature ($T_{mid}$), column effective extinction coefficient ($\sigma_{ext}$) and ice water content (IWC)) and net radiative forcings of the cirrus cloud measured by Micro Pulse Lidar at the Barcelona lidar station, by CALIPSO satellite (Gil-Díaz et al., 2024) and with the properties obtained with CLaMS-Ice, at bottom-of-the-atmosphere (BOA), top-of-the-atmosphere (TOA) and in the full atmosphere (ATM). [a]The value of ice water content is calculated with the self-consistent scattering model for cirrus cloud (see Sec. 3.1).[b] The CLaMS-Ice product does not provide a value of the effective column extinction of the cirrus cloud.

Taking the new methodology explained above to characterize radiatively cirrus clouds, the self-consistent scattering model for cirrus clouds is used to obtain the effective column extinction coefficient, single scattering albedo and asymmetry factor to introduce them into the ARTDECO package and calculate its net radiative forcing. In the case of cirrus cloud detection with

MPL measurements, as described above, the mid-cloud temperature and the effective column extinction coefficient, obtained by the two-way transmittance method (Gil-Díaz et al., 2024), are employed to characterize the cirrus radiatively. Alternatively, as the CALIPSO and CLaMS-Ice data provide an IWC measurement for such cirrus cloud, this data together with the mid-cloud temperature is used to represent the cirrus radiatively. Despite the fact that CALIPSO passes 78 km from the Barcelona station, in the simulations with these data and those of CLaMS-Ice, an albedo and surface temperature corresponding to the

Barcelona lidar station are considered, in order to compare radiative forcing values.

In Table 2 one observes that the properties of the cirrus cloud are similar, except for the IWC, which is considerably higher for the cirrus characterized with CALIPSO measurements. If the IWC is calculated using the extinction coefficient and mid-





cloud temperature values provided by CALIPSO, the ensemble scattering model for cirrus clouds yields an IWC value of 3.9

mg/m$^3$. This result is significantly lower than the IWC measured by CALIPSO, since the ensemble scattering model for cirrus clouds often underestimates the IWC of cirrus clouds measured at mid-latitude (Baran and Labonnote, 2007). So a certain discordance between the net radiative forcing magnitudes would be expected. For all three simulations, the cirrus cloud warms both at TOA and at BOA, since due to the time (2 UTC) the solar radiation component is null. Moreover, as the forcing is proportional to the COD, the cirrus characterized with the CLaMS-Ice products produces a slightly higher forcing than with

the other data. On average, it can be established that the cirrus cloud measured at Barcelona lidar station produces a warming at BOA of +19.19±2.08 Wm$^{-2}$, at TOA of +49.07±5.71 Wm$^{-2}$ and in the full atmosphere of +29.88±3.63 Wm$^{-2}$.

## 5   Conclusions

In this paper a study of radiative properties and forcings of cirrus clouds based on 4 years of continuous ground-based lidar measurements with the Barcelona (Spain) Micro Pulse Lidar (MPL) is analysed. First, a new approach of a self-consistent

scattering model for cirrus clouds is presented to get the radiative properties of cirrus clouds at different wavelengths with only the effective column extinction coefficient calculated with the two-way transmittance method and mid-cloud temperature, from radiosounding data. The self-consistent scattering model for cirrus clouds consists of an ensemble of six ice crystal members, where the simplest ice crystals are represented by hexagonal ice columns with an aspect ratio of unity and the more complex ice crystals are formed by arbitrarily and randomly oriented attaching other hexagonal elements, until a chain-like ice

crystal is formed. The members of the ensemble are distributed into six equal intervals of the particle size distribution (PSD), which relates the decimal logarithm of ice water content (LIWC) to a radiative property (scattering, absorption coefficient and asymmetry factor) via a polynomial fit to the in-cloud temperature (T), with a spectral dependence. The new approach for the calculation of the radiative properties of cirrus clouds at different wavelengths consists of first calculating the IWC of the cirrus cloud by introducing the effective column extinction coefficient in an equation derived from the model, for

a mid-cloud temperature. This equation is obtained by assuming the absence the absorption, which is entirely reasonable because the wavelength used belongs to the visible spectral range. Once, the IWC is estimated, this variable is introduced again into the model to get the absorption, scattering and asymmetry factor coefficients for each wavelength, respectively. Applying this method to cirrus clouds measured in Barcelona during November 2018 to September 2022 at 00 and 12 UTC, it is obtained that the average of the ice water content is 4.97±5.53 mg/m$^3$, the single scattering albedo is 0.99±0.05 and

the asymmetry factor is 0.76±0.00 at 0.55 $\mu$m. Second, the radiative forcing of cirrus clouds is calculated with the radiative transfer model DISORT. Radiative fluxes are validated at bottom-of-the-atmosphere with SolRad-Net pyranometers in the shortwave spectral range, and at top-of-the-atmosphere with CERES measurements in the longwave spectral range. One one hand, most downward radiative fluxes calculated with the DISORT model overestimate the SolRad-Net observations, resulting in a BIAS of +11.61% and a slope of the linear regression (a = 0.71±0.20). On the other hand, a large difference in upward

radiative fluxes between simulated and observations is found for each cirrus cloud scene, resulting in a BIAS of +51.62%. Third, a validation of the importance of the planetary boundary layer aerosols in the cirrus scenes simulations is carried out.



Calculations with and without aerosols of the cirrus direct radiative forcings are made to assess the error induced by neglecting tropospheric aerosols, which results in a negligible BIAS of -0.71%. In the other case where the aerosols are vertically closer to the clouds, the simplification of not considering aerosols in the calculation of cloud forcings may not be valid, leading to a significant underestimation of cloud forcings. Forth, the radiative forcings of cirrus clouds are calculated distinguishing between nighttime and daytime. At nighttime, cirrus clouds warm the atmosphere with radiative effects at TOA almost double than at BOA, with the thicker cirrus clouds contributing most to the forcing. At daytime, cirrus clouds generally cool the BOA (80% of the cases) and always warm the TOA, resulting in a warming of the full atmosphere. On average, at nighttime, cirrus clouds warm $+23.02\pm21.23$ Wm$^{-2}$ at BOA, $+50.06\pm42.99$ Wm$^{-2}$ at TOA and in the full atmosphere $+27.04\pm22.39$ Wm$^{-2}$; at daytime, cirrus clouds cool $-8.57\pm14.95$ Wm$^{-2}$ at BOA and warm $+18.94\pm16.95$ Wm$^{-2}$ at TOA, and in the full atmosphere $+27.51\pm26.63$ Wm$^{-2}$. Fifth, the variation of the cirrus cloud radiative forcing at daytime is also analysed as a function of the SZA: it shows that at TOA for the real cases the average net radiative forcing is always positive and for the results from a sensitivity study, the mean net radiative forcing shifts from positive to negative values at 58º. For all cases, a slight downward tendency of the net radiative forcing is also found. At BOA, most net radiative forcing values are negative. The change from positive to negative values of the mean net radiative forcing is only observed for the real cases, being SZA crossover of 25º. Sixth, for a case study, the radiative forcing of a cirrus cloud along its back-trajectory is analysed using CLaMS-Ice products. During the overlap between the back-trajectory of this cirrus cloud and the CALIPSO satellite on 11/02/2019 02:03:20 UTC, a brief validation of the IWC is made, resulting in a relative error of 29% for the CLaMS-Ice output. This cirrus cloud comes from the Atlantic Ocean, passing through part of France and Barcelona to reach the Mediterranean Sea. Over the Atlantic Ocean, the air mass which corresponds to the cloud undergoes a rise in height, reaching a minimum in temperature and a COD of 0.6. After this initial cirrus cloud fast growth, the cloud gradually fades away as it approaches France, where the CLaMS-Ice model gives a null IWC. Afterwards, the cirrus cloud begins to form again on land surface and to grow up. Along its trajectory, the cirrus cloud produces a net radiative forcing that goes at TOA from values close to 0 to $+42$ Wm$^{-2}$, at BOA from negative values, whose minimum is $-51$ Wm$^{-2}$ to positive values reaching a maximum of $+20$ Wm$^{-2}$ and in the full atmosphere varying between values close to 0 to $+48$ Wm$^{-2}$, being the maximum. Finally, the radiative forcings that the cirrus cloud has at the beginning of the back-trajectory are compared using different measurements (MPL and CALIPSO measurements and CLaMS-Ice outputs) and making use of the self-consistent scattering model for cirrus clouds. It results in an average warming at BOA of $+19.19\pm2.08$ Wm$^{-2}$, at TOA of $+49.07\pm5.71$ Wm$^{-2}$ and in the full atmosphere of $+29.88\pm3.63$ Wm$^{-2}$.

*Data availability.* The MPLNET products are publicly available on the MPLNET website (https://mplnet.gsfc.nasa.gov/download_tool/) (MPLNET, 2024) in accordance with the data policy statement. Radiosoundings data are available upon request from the authors or Meteocat. The SolRad-Net product is publicly available on the SolRad-Net website (https://solrad-net.gsfc.nasa.gov/) (Goddard Space Flight Center, 2024a) in accordance with the data policy statement. The CERES products are publicly available on the CERES website (https://ceres.larc. nasa.gov/) (Langley Research Center, 2024). The AERONET products are provided by a federation of ground-based remote sensing aerosol networks established by NASA and PHOTONS (PHOtométrie pour le Traitement Opérationnel de Normalisation Satellitaire) and is greatly



expanded by collaborators from national agencies, institutes, universities, individual scientists, and partners. The AERONET products are publicly available on the AERONET website (https://aeronet.gsfc.nasa.gov/) (Goddard Space Flight Center, 2024b). The CALIPSO product is provided by the NASA Langley Research Center's (LaRC) ASDC DAAC and is managed by the NASA Earth Science Data and Information System (ESDIS) project. NASA data are freely accessible and available on the Atmospheric Science Data Center website (https://asdc.larc.nasa.gov/) (NASA, 2024).

*Author contributions.* CGD prepared the automatic algorithm to get the radiative properties and forcings of cirrus clouds for MPL and radiosounding data. CGD prepared the figures of the paper. MS, OS, AS, CMP, AC, ARG and DCFSO reviewed different parts of the results. DCFSO also took care of the maintenance of the MPL. CGD prepared the paper, with contributions from all co-authors.

*Competing interests.* At least one of the (co-)authors is a member of the editorial board of Atmospheric Chemistry and Physics.

*Acknowledgements.* The CLaMS-Ice model is based at Forschungszentrum Jülich, ICE-4, Germany, and has been used with their kind
permission. The authors acknowledge the support of the ACTRIS European Research Infrastructure Consortium (ERIC). The first author would like to express her gratitude to all the members of the Laboratoire d'Optique Atmosphérique (LOA), and especially to Philippe Dubuisson and Odran Sourdeval, for their welcome and hospitality.

*Financial support.* This research has been partly funded by the Spanish Agencia Estatal de Investigación (grant no. PID2019-103886RB-I00) and the European Commission through the Horizon 2020 Programme (project ACTRIS IMP, grant agreement
no. 871115; ATMO-ACCESS, grant agreement no. 101008004; GRASP-ACE, grant agreement no. 778349) and through the Horizon Europe Programme (project REALISTIC, grant agreement no. 101086690).





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
