# Peer review of "Study of radiative properties and effects of high-altitude cirrus clouds in Barcelona, Spain with 4 years of lidar measurements"

_EGUsphere, 2024_

## Author Comment (AC1)

Dear Reviewer,

I attach in this document the answers to your comments. But first of all, I would like to thank you for spending time with the review of this manuscript. The answers are in blue and a new manuscript has been created to visualize the changes, with new contributions in red and deleted contributions in strikeout.

**General comments:**

The topic of this paper is very interesting and within the scope of ACP. The results presented in Section 4 often differ from expectations or from findings reported in the literature. In my opinion, **the discussion tends to be too vague and should be improved and detailed before the paper can be considered for publication.**

Some clarifications are needed in Section 3 about the Methodology, as will be detailed hereafter.

Some portions of the text in Sections 2 and 3 are copied from websites or from published papers. The authors should use their own words and, if not possible, clearly quote the original work.

Thank you for your feedback. We have carefully revised Sections 2 and 3, rewriting the relevant paragraphs to ensure they are properly paraphrased. We think these changes improve the overall clarity of the manuscript.

**Specific comments:**

Section 3.1:

1) The presentation of the self-consistent scattering model for cirrus clouds is confusing and the contribution from the authors is difficult to identify. Most of the text between lines 154 to 184 is directly copied from other publications (which should be properly indicated using quotations). The authors introduce Equations (1) and (2) as "moment parameterization of the PSD by Field et al. (2007)" (line 170), but I think that there is a confusion between PSD parameterization (which has IWC and T as input) and the parameterizations shown in Equations (1) and (2). Equations (1) and (2) are noted "Baran model" in Fig.1. They happen to be very similar to those presented by Vidot et al. (2015, J. Geophys. Res. Atmos., 120, 6937–6951, doi:10.1002/2015JD023462), which are not cited in the paper. Please clarify how Equations (1) and (2) were established.

The description of self-consistent scattering model for cirrus clouds together with Figure 1 have been changed to make them clearer. The authors apologize because they were not aware of the (Vidot et al., 2015) study. Indeed, equations (1) and (2) correspond to (Vidot et al., 2015) equations (2) - (4). Therefore, (Vidot et al., 2015) has been quoted and these expressions have been omitted in this manuscript.

2) Please clarify whether Equation (3) has always one unique physical solution.

Right, the Eq. 3 always has one unique physical solution and it is achieved with the addition operator. Therefore, the ± operator has been changed to +, in order to avoid confusion. The paragraph (177-179 lines) with the new changes is shown as follows:

"This formulation provides a unique physical solution and simplifies the IWC calculation based on extinction coefficient and cloud temperature, assuming no absorption, which is entirely reasonable because the working wavelength lies within the visible spectral range (Sun and Shine, 1994)."

**3)** The notion of effective column extinction coefficient is introduced line 188, but is not defined until lines 307-309 in Section 4.1. Can you explain why you need to introduce this effective extinction as a value representative of the entire cirrus? Is there a need to have one single set of parameters over a smaller geometric thickness for the radiative transfer calculations? My understanding is that COD is unchanged and that the effective extinction is twice the mean extinction. Can you clarify? How does this change of visible extinction coefficient affect IWC, other extinction coefficients, SSA, and asymmetry parameter? In Section 4.1 (Fig. 3), you indicate that your average IWC of 5 mg m$^{-3}$ is close to a value of 3 mg m$^{-3}$, which is in agreement with other studies. Does this suggest that mean extinction would be better suited than effective extinction?

After applying the two-way transmittance method, we have vertical profiles of extinction and temperature for each cirrus cloud, with a vertical resolution of 75 m. Before applying the self-consistent scattering model for cirrus clouds, we degrade the vertical resolution of these profiles to constrain them to the model vertical resolution, through vertical averaging. This vertical resampling of the cloud is necessary to be able to distribute it in different model layers and not to consider the cirrus cloud in a single 1 km thick layer.

Figure 3 shows the mean values for each cloud. To avoid confusion, the term effective column extinction coefficient has been changed to extinction coefficient in each vertical layer of the model and in Figure 3 it is specified that these are the mean values of each cirrus cloud. That the mean IWC is 5 mg m$^{-3}$, being a close value of 3 mg m$^{-3}$, which is in agreement with other studies means that 1) the new methodology for calculating the optical scattering properties of cirrus clouds adequately estimates the IWC, 2) the new approach of the two-way transmittance method inverts well the lidar signal and 3) the identification of cirrus clouds made with measurements from the MPL at the Barcelona lidar station is correctly.

The vertical resampling of the cirrus cloud in the self-consistent scattering model could sometimes lead to a cloud layer having a low ice water content and consequently low optical scattering values, as mentioned in lines 298-305.

**4)** Can you please give T and IWC for the cloud used to create Fig. 2?

Figure 2 has been achieved by averaging the optical scattering properties in all the layers of a cirrus cloud measured on 8$^{th}$ December 2018 at 12 UTC in Barcelona lidar station. The figure with the vertical distribution of the optical scattering properties is shown below.

[Figure]

The vertical profiles of the extinction, temperature and ice water content of this cirrus cloud are shown below.

[Figure]

Section 3.2:

**5)** Line 240: where in the paper is the "with gases only" configuration used?

This configuration is used to calculate the direct radiative effect of the cirrus cloud as the difference between the radiative fluxes of the simulations in which cirrus clouds and gases and only gases are considered. This direct radiative effect has been defined as $DRE_{GC-G}$. I recommend reviewing Section 4.3, where these direct radiative effects are analyzed.

**6)** Lines 246-257: do you mean that these properties are used as inputs to the model (i.e. not parameterized)?

The extinction, single scattering albedo and asymmetry factor obtained from the self-consistent scattering model for cirrus clouds are used as inputs to the RTM model. In accordance with your suggestion to replace radiative properties by optical scattering properties, this paragraph (231-232 lines) has been rewritten as follows:

"The cirrus clouds are parameterized in the RTM model geometrically and optically, with the results obtained (Gil-Díaz et al., 2024) and the retrievals obtained with the self-consistent scattering model for cirrus clouds (see Section 3.1)."

**7)** Surface properties (section 3.2.2): surface temperature is from the CERES product and is averaged monthly. Please elaborate. Surface temperature might exhibit large daily variations over land, and I wonder 1) about temporal mismatches between CERES observations at 10-10:30 UTC and 12-13 UTC and the ground-based observations in Barcelona, and 2) about the variations within a given month. Errors in surface temperature could introduce errors in the LW radiative transfer calculations. Did you use monthly mean surface temperatures for the comparisons with CERES in Fig. 4?

Yes, we used monthly surface temperatures to compare with CERES observations. The albedo and temperature variations although small, you are right that they induce an error in the calculation of radiative fluxes in the longwave spectrum. Thank you very much for this comment because we had not realized that a large part of the problem was the time averaging of surface properties. To address this, we have downloaded new data on upward radiative fluxes at top-of-the-atmosphere, along with the surface emissivity, surface temperature and cloud mask from the NOAA-20 satellite to provide a temporal coincidence between satellite and ground-based observations over Barcelona. We then incorporated instantaneous values for surface albedo and temperature into the simulations, which were rerun accordingly. The results obtained from the comparison of the upward radiative fluxes in the longwave spectrum at top-of-the-atmosphere are shown in the figure below.

[Figure]

As you can see, the BIAS has decreased to +33.6% and the points exhibit a more linear trend. Despite the drop in BIAS, the current value is still considerable. As discussed in the manuscript, NOAA-20 may discern a different atmospheric scene, even covering part of the Mediterranean Sea. Further analysis of the cloud mask reveals that 14% of the cases analyzed have more than 90% of clear sky footprint area, highlighting the complexity of the comparison between simulations and satellite observations at the top-of-the-atmosphere. The full discussion can be found in lines 326-341.

In any case, given the significant improvement in validation with NOAA-20 data, we proceeded to rerun all the simulations to continue analyzing the direct radiative effect of cirrus clouds. The previous results in the manuscript have been replaced with those obtained with these new simulations.

**Section 4**

**8)** Section 4.1, Fig. 3; can you please show the IWC values smaller than 1 mg m$^{-3}$?

Yes, of course. In the probability histogram of the ice water content, the logarithmic scale on the X-axis has been changed to a normal one, with a bin size of 2.5 mg m$^{-3}$. I show you the new figure below:

[Figure]

If a bin size of 1 mg m$^{-3}$ is considered, the histogram looks a bit uglier, which is why we have preferred to show the figure above. Nevertheless, in this document I show you this additional figure below (figure on the left) with a zoom (figure on the right) for cases with IWC less than 1 mg m$^{-3}$.

[Figure]

9) Section 4.2, lines 340-342: I do not understand the reasoning: the authors first derive IWC from the lidar visible extinction coefficient and then derive all the other properties. Can you explain why the issue is related to small IWCs? I am wondering how the asymmetry parameter could play a role. Simple sensitivity studies could strengthen the discussion.

We derive the IWC from the extinction coefficient obtained with an elastic lidar signal from a MPL because there is no ground-based instrument available at the Barcelona lidar station to directly measure IWC.

The issue of small values of IWC lies in the degradation of the cirrus vertical distribution when is constrained to the model vertical resolution. Initially, the vertical extinction profile has a resolution of 75 m and is converted into a vertical profile of 1 km resolution by averaging. Cirrus clouds are not uniformly

distributed over the entire 1 km layer. As a result, in some vertical layers of the model a low extinction value is obtained and after applying Eq. 3, these layers have a low IWC estimation.

Additionally, as you suggested, a sensitivity study was done, but due to the length of the manuscript, it has not been included. Below, I present the optical scattering properties derived from the self-consistent scattering model for cirrus clouds at 0.55 µm, corresponding to initial extinction values between 0 and 2 Mm$^{-1}$ at a temperature of -37ºC. By initial extinction values I mean the extinction values entered in Eq. 3 to obtain the IWC. The results are shown below.

[Figure]

As can be seen in the figure above, for initial extinction values lower than 1 Mm$^{-1}$, the absorption is not negligible at 0.55 μm. Consequently, in the 3 cases mentioned in the manuscript (lines 299-304), the averaging of the SSA in all vertical layers results in a value lower than 0.9.

**10)** Section 4.2, lines 345 – 359 and Fig. 4 (right): Collocation issues could indeed be the reason for the very large discrepancies, but I suggest more detailed discussions. For the SVC samples in red, the results are similar to cloud free simulations, which are very sensitive to surface temperature and emissivity. All the red samples have very similar simulated values, which suggests very similar surface parameters. Did you use seasonal (surface emissivity) and monthly (surface temperature) means for these comparisons? If yes, would the comparisons be improved using the CERES parameters reported for each individual case?

Yes, we used monthly surface temperatures and seasonal surface emissivity to compare with CERES observations. The reanalysis made has been discussed in question 7).

**11)** Section 4.4: TOA net DRF during daytime (Fig. 6 top right) is always positive for COD up to 1.2. This is in contradiction with previous studies such as for instance Campbell et al. (2016) who find positive net DRF until COD = about 0.37-0.56 and negative DRF for larger CODs. Indeed, "multiple factors are involved (line 402)", but some discussion could be added. I note that TOA LW DRE of Figure 6 is much larger than in Fig. 2 of Campbell et al., 2016 for a given COD, while the TOA SW DRE are similar in both figures. What could cause larger TOA LW DRF in this study? Could it be related to a larger difference between surface and cloud temperature? Can you clarify here how surface temperature was determined and give values?

Right, all your comments have been added to the discussion. The paragraph with the new comments is shown below.

"At daytime, at TOA, the net direct radiative effect remains positive for all cirrus clouds, dominating the positive longwave component. This effect has been observed in (Campbell et al., 2016) for COD up to approximately 0.6. For higher COD values, (Campbell et al., 2016) reports a negative NET TOA DRE. In this study, a decreasing trend in NET TOA DRE is observed from COD values of 0.5, although no negative values are obtained. Additionally, the LW NET TOA DRE component grows faster than the one reported by (Campbell et al., 2016), suggesting that negative values of NET TOA DRE could occur for cirrus clouds with higher COD than those found in (Campbell et al., 2016). This discrepancy may be due to the higher surface emissivity and temperature values considered in the present work. Further measurements of NET TOA DRE for cirrus clouds with higher COD are needed to confirm the decreasing trend. In contrast, at BOA, the net direct radiative effect is almost always negative (only 18% of the cases show a positive net direct radiative effect, whose value is close to 0), being the outgoing shortwave radiation in the presence of cirrus clouds larger than in cirrus cloud free conditions. The albedo effect overcomes the greenhouse effects in the SW range because of low absorption capacity of the small crystals, as shown in Fig. 2. These changes of sign which occur for thin cirrus clouds, like in other studies (Campbell et al., 2016; Lolli et al., 2017; Kramer et al., 2020), where the dominant factor in the change of sign of forcing is unclear. Multiple factors are involved, from the optical properties of the cirrus, to the solar zenith angle or the surface temperature and surface albedo (Wolf et al., 2023)."

**12)** Section 4.5: Figure 8 shows only net DRFs, and it is therefore difficult to assess whether the difference between the sensitivity study and the real cases is influenced by LW or SW. Assuming that differences are due at least in part to the LW, I wonder how the surface temperature = 28 ºC and the longwave

surface albedo used in the simulations compare with the real case values. Can you identify which parameters cause the oscillations in the black curve corresponding to the real cases?

As mentioned in lines 445-449, the high variability of the net direct radiative effect for the real cases could be explained by the different values of cloud optical depth, cirrus optical scattering properties, surface albedo and temperature considered in the simulations. An in-depth sensitivity analysis should be done to analyze the weights of each variable, but it is observed that the cloud optical depth is the variable that could have the most weight since for each averaged SZA value, the difference between the mean CODs is very large. Re-running the simulations with updated surface albedo and temperature values has also shown that the oscillations in the black curve for the real cases have been somewhat attenuated, though not completely eliminated. This finding further supports the idea that the oscillations are mainly caused by variations in COD.

**13)** Section 4.6: the daytime net TOA DRF is close to zero when the cloud is over oceans. What are the surface parameters (which values)? Would the same analysis but using the same surface parameters as in Barcelona yield positive differences as found earlier in the paper?

Possibly. Due to improvements in the evaluation of radiative fluxes in the longwave spectrum at top-of-the-atmosphere, new shortwave surface albedo, longwave emissivity and surface temperature data from the NOAA-20 satellite have been downloaded. The simulations were re-run using instantaneous surface property values. The figure below shows the new results.

[Figure]

As can be seen in the figure above, the direct radiative effect of the cirrus cloud along its back-trajectory is very similar but slightly lower. The NET TOA DRE over the Atlantic Ocean remains very small, practically negligible. The surface properties considered in the new simulations are shown in the figure below.

[Figure]

**Other comments:**

- Asymmetry Factor (asyF) is traditionally called asymmetry parameter and noted g. Please consider changing these notations.
- I would call Single scattering albedo (SSA) and asymmetry parameter "optical scattering properties" rather than "radiative properties".

The authors accept your suggestions and have changed the manuscript accordingly.

---

## Author Comment (AC2)

Dear Reviewer,

I attach in this document the answers to your comments. But first of all, I would like to thank you for spending time with the review of this manuscript. The answers are in blue and a new manuscript has been created to visualize the changes, with new contributions in red and deleted contributions in strikeout.

**Major comments**:

**1. Study Design and Methodology**

1. **Self-Consistent Scattering Model**:
   o The **description of the scattering model** (Section 3.1) lacks clarity, and the contribution of the authors versus existing models is unclear.

   o The equations used (1–3) need clearer documentation on their derivation. For example, the similarity of Equations (1) and (2) to Vidot et al. (2015) suggests that a citation or discussion is necessary.

   o **Equation (3)**: Clarify whether it has a unique physical solution under all conditions.

The description of self-consistent scattering model for cirrus clouds together with Figure 1 have been changed to make them clearer. The authors apologize because they were not aware of the (Vidot et al., 2015) study. Indeed, equations (1) and (2) correspond to (Vidot et al., 2015) equations (2) - (4). Therefore, (Vidot et al., 2015) has been quoted and these expressions have been omitted in this manuscript.

It has also been clarified that the Eq. 3 always has one unique physical solution and it is achieved with the addition operator. Therefore, the ± operator has been changed to +, in order to avoid confusion. The paragraph (177-179 lines) with the new changes is shown as follows:

"This formulation provides a unique physical solution and simplifies the IWC calculation based on extinction coefficient and cloud temperature, assuming no absorption, which is entirely reasonable because the working wavelength lies within the visible spectral range (Sun and Shine, 1994)."

2. **Parameterization Choices**:
   o The use of **effective column extinction coefficient** to simplify radiative transfer modeling needs more justification. Explain how this affects IWC, single scattering albedo (SSA), and the asymmetry parameter.

To avoid confusion, the term effective column extinction coefficient has been changed to extinction coefficient in each vertical layer of the model. The resampling of the cloud extinction is necessary because after applying the two-way transmittance method, a vertical extinction profile with a vertical resolution of 75 m is obtained and needs to be rescaled to the model vertical resolution of 1 km.

The vertical resampling of the cirrus cloud extinction in the self-consistent scattering model could sometimes lead to a cloud layer having a low ice water content and consequently low optical scattering values, as mentioned in lines 298-305.

3. **Surface Properties**:
   o The use of **monthly averaged surface temperatures** from CERES could introduce biases in longwave flux calculations, particularly given daily variations in land temperatures. A discussion of this limitation and exploration of case-specific surface temperature/emissivity values are needed.

Yes, we used monthly surface temperatures to compare with CERES observations. Although the albedo and temperature variations are small, you are right that they induce a bias in the calculation of radiative fluxes in the longwave spectrum. To address this, we have downloaded new data on upward radiative fluxes at top-of-the-atmosphere, along with the surface emissivity, surface temperature and cloud mask from the NOAA-20 satellite to provide a temporal coincidence between satellite and ground-based observations over Barcelona. We then incorporated instantaneous values for surface albedo and temperature into the simulations, which were rerun accordingly. The results obtained from the comparison of the upward radiative fluxes in the longwave spectrum at top-of-the-atmosphere are shown in the figure below.

[Figure]

As you can see, the BIAS has decreased to +33.6% and the points exhibit a more linear trend. Despite the drop in BIAS, the current value is still considerable. As discussed in the manuscript, NOAA-20 may discern a different atmospheric scene, even covering part of the Mediterranean Sea. Further analysis of the cloud mask reveals that the 14% of the cases analyzed have more than 90% of clear sky footprint area, highlighting the complexity of the comparison between simulations and satellite observations at the top-of-the-atmosphere. The full discussion can be found in lines 326-341.

Given the significant improvement in validation with NOAA-20 data, we proceeded to rerun all the simulations to continue analyzing the direct radiative effect of cirrus clouds. The previous results in the manuscript have been replaced with those obtained with these new simulations.

4. **Limited Validation of IWC**:
   o While lidar extinction profiles are used to derive IWC, the validation of these IWC values against independent datasets is limited. For instance, thin cirrus clouds may have underestimated IWC, affecting radiative forcing results. More references and few lines of discussion are needed.

Figure 3 shows the mean IWC distribution for each cirrus cloud scene and lines 291-296 compare the values with the literature. Further references have been added based on your suggestion. The paragraph is shown as follows:

"In Fig. 3 one observes that cirrus clouds have an IWC between 0.03 and 30 mg m$^{-3}$, being characteristic of mid-latitude cirrus clouds (Korolev et al., 2001; Field et al., 2005, 2006; Schiller et al., 2008; Baran et al, 2011b; Sourdeval, 2012; Kramer et al.,2016, 2020). Where the average of IWC is ~5 mg m$^{-3}$, being a value close to 3 mg m$^{-3}$, which is the central value of the mid-latitude ice cloud distributions obtained by (Sourdeval, 2012) and the mean value of IWC for temperatures between 210 and 235K found in (Kramer et al., 2016). A slightly higher measured IWC value of 7 mg m$^{-3}$ was found by (Korolev et al., 2001) for cirrus clouds whose temperature ranged from 233 to 243K."

**2. Results and Data Analysis**

1. **Large Bias in Longwave Fluxes**:
   o The **+51% bias** in simulated longwave fluxes at TOA compared to CERES observations is significant. While collocation issues are suggested as the cause, this explanation is not robust. Other potential causes, such as surface temperature/emissivity inaccuracies or errors in IWC parameterization, should be explored and stated in the text.

As mentioned in question 3, with the new NOAA-20 satellite instantaneous surface temperature and albedo data, it has been possible to reduce the BIAS of the longwave upward fluxes simulated and observed to +33.6%. In addition, as discussed in the lines 333-341, other possible causes of error may be involved in having such a BIAS, such as the satellite observing a different atmospheric scene or covering part of the Mediterranean Sea. The entire paragraph is shown below.

"In our case, the large BIAS = +33.6 % obtained could be due to the spatial resolution of the observed measurements taken, which may cover part of the Mediterranean Sea. In addition, the cloud mask associated with each observation indicates that in 14% of the cases it has more than 90% of clear sky footprint area. As demonstrated by (Gil-Díaz et al., 2024) most the cirrus clouds are visible and their horizontal extension is smaller than the cirrus clouds that form at higher altitudes (well-known as sub-visible cirrus clouds) (Kramer et al., 2020). This makes them more challenging to detect from top-of-the-atmosphere. Hence, the comparison of simulated radiative fluxes and NOAA-20 observations is not as trivial and conclusive as with SolRad-Net observations, since the NOAA-20 satellite can observe a slightly different atmospheric situation, as mentioned above. Not to mention the limitations of the 1-D radiative transfer model DISORT to represent an irregular composition of broken and/or overlapping clouds that the NOAA-20 satellite could observe."

2. **Daytime TOA Net Radiative Forcing**:
   o The finding that **net daytime TOA forcing remains positive for COD < 1.2** (Section 4.4) contrasts with prior studies. For example, Campbell et al. (2016) report a transition to negative forcing for COD > 0.7 (for 30sr solution). A broader discussion on the discrepancies from Campbell et al. (2016) is required.

Right, this discrepancy has been highlighted in this paper and discussed in lines 381-389. The full paragraph is shown below:

"At daytime, at TOA, the net direct radiative effect remains positive for all cirrus clouds, dominating the positive longwave component. This effect has been observed in (Campbell et al., 2016) for COD up to approximately 0.6. For higher COD values, (Campbell et al., 2016) reports a negative NET TOA DRE. In this study, a decreasing trend in NET TOA DRE is observed from COD values of 0.5, although no negative values are obtained. Additionally, the LW NET TOA DRE component grows faster than the one reported by (Campbell et al., 2016), suggesting that negative values of NET TOA DRE could occur for cirrus clouds with higher COD than those found in (Campbell et al., 2016). This discrepancy may be due to the higher surface emissivity and temperature values considered in the present work. Further measurements of NET TOA DRE for cirrus clouds with higher COD are needed to confirm the decreasing trend."

3. **Back-Trajectory Analysis**:
   o The investigation of cirrus cloud radiative properties using the CLaMS-Ice model and CALIPSO is interesting but underdeveloped. The daytime net TOA forcing close to zero over oceans raises questions about the surface parameters used.

Due to improvements in the evaluation of radiative flux in the longwave spectrum at top-of-the-atmosphere, new shortwave surface albedo, longwave emissivity and surface temperature data from the NOAA-20 satellite have been downloaded. The simulations were re-run using instantaneous surface property values. The figure below shows the new results.

[Figure]

As can be seen in the figure above, the direct radiative effect of the cirrus cloud along its back-trajectory is very similar but slightly lower. The NET TOA DRE over the Atlantic Ocean remains very small, practically negligible. The surface properties considered in the new simulations are shown in the figure below.

[Figure]

**3. Writing Quality**

1. **Copied Text**:
   o Sections 2 and 3 contain text copied from external sources. These sections should be rephrased in the authors' own words or explicitly quoted with citations.

Thank you for your feedback. We have carefully revised Sections 2 and 3, rewriting the relevant paragraphs to ensure they are properly paraphrased. We think these changes improve the overall clarity of the manuscript.

2. **Clarity and Conciseness**:
   o The discussion sections tend to be vague and require more **precise explanations**. For example, the reasoning behind discrepancies in longwave fluxes **(Fig. 4)** and **TOA net forcing (Fig. 6)** should be better supported by sensitivity studies or external validation.

Thank you for your feedback. We have carefully revised the explanations of the discrepancies in the evaluation of the longwave radiative fluxes observed and simulated at TOA and the positive direct radiative effect at TOA for all COD.

**4. Literature Review and Comparison**

1. **Novelty Discussion**:

- The introduction of the inverse Baran model is claimed as novel. However, similar approaches have been explored in other studies. The manuscript should more clearly articulate its novelty compared to prior work.

In order to improve this part of the methodology, (Vidot et al., 2015) has been cited and the equations of the self-scattering model for cirrus clouds have been omitted. I have also tried to highlight the authors' contribution to the developed methodology based on the self-scattering model for cirrus clouds.